# IL-26 from innate lymphoid cells regulates early-life gut epithelial homeostasis by shaping microbiota composition

Yazan Salloum [ID][1], Gwendoline Gros[1], Keinis Quintero-Castillo [ID][2], Camila Garcia-Baudino[1], Soraya Rabahi[1], Akshai Janardhana Kurup[1], Patricia Diabangouaya[1], David Pérez-Pascual[3], Rodrigo A Morales Castro [ID][4,5], Jos Boekhorst[6], Eduardo J Villablanca[4], Jean-Marc Ghigo [ID][3], Carmen G Feijoo [ID][2], Sylvia Brugman [ID][6] & Pedro P Hernandez [ID][1✉]

## Abstract

**Animals host symbiotic microbial communities that shape gut health. However, how the host immune system and microbiota interact to regulate epithelial homeostasis, particularly during early development, remains largely unclear. Human interleukin-26 (IL-26) is associated with gut inflammation and has intrinsic bactericidal activity in vitro, yet its in vivo functions are largely unknown, primarily due to its absence in rodents. To examine the role of IL-26 in early life, we used zebrafish and found that gut epithelial cells in *il26-/-* larvae exhibited increased proliferation, faster turnover, elevated DNA damage, and altered cell population abundance. This epithelial dysregulation occurred independently of the IL-26 canonical receptor and resulted from dysbiosis in *il26-/-* larvae. Moreover, IL-26 bactericidal activity was conserved in zebrafish, suggesting a potential role of this property in regulating microbiota composition. We further identified innate lymphoid cells (ILCs) as the primary source of IL-26 at this developmental stage. These findings establish IL-26 as a central player in a regulatory circuit linking the microbiota, ILCs, and intestinal epithelial cells to maintain gut homeostasis during early life.**

**Keywords** Cytokines; DNA Damage; Innate Lymphoid Cells; Proliferation; Zebrafish
**Subject Categories** Development; Immunology; Microbiology, Virology & Host Pathogen Interaction

See also: K Wright & S Mostowy

## Introduction

The gastrointestinal tract is a complex system housing numerous populations of immune cells and commensal microorganisms. However, it is also a port of entry for various pathogens. During early life, a dynamic crosstalk between immune cells, epithelial cells, and colonizing microbiota is critical for establishing life-long intestinal homeostasis (Zheng et al, 2020; Paucar Iza and Brown, 2024; Al Nabhani et al, 2019). Dysregulation of this crosstalk can lead to inflammatory bowel disease (IBD), a group of disorders closely associated with alterations in the gut microbiota that can elicit a chronic immune response and lead to tissue damage. In addition, IBD patients are at higher risk of developing colon cancer, a hallmark of which is greater genomic instability and increased proliferation of abnormal cells (Zhou et al, 2023). Therefore, identifying factors that regulate early-life microbiota composition, epithelial cell proliferation, and DNA damage is pivotal for advancing early diagnosis, prevention, and treatment of intestinal pathologies.

Genome-wide association studies have identified interleukin-26 (IL-26) as a risk locus for ulcerative colitis (UC) (Silverberg et al, 2009), indicating a potential role of this cytokine in intestinal homeostasis. Moreover, IL-26 has been found to be overexpressed in the inflamed mucosa of IBD patients (Song et al, 2022; Dambacher et al, 2009). Notably, intestinal epithelial cells (IECs) were shown to express the IL-26 receptor complex (IL10RB and IL20RA) and to respond to this cytokine in vitro (Dambacher et al, 2009). Despite these findings, the in vivo functions of IL-26 and its impact on IECs remain largely unclear, primarily due to the absence of this cytokine in rodent models (Donnelly et al, 2010).

The human IL-26 protein has been demonstrated to exhibit receptor-independent functions. For example, it has been shown that IL-26 can form complexes with DNA and activate intracellular TLR9 in dendritic cells (Meller et al, 2015). Furthermore, in vitro

[1]Institut Curie, PSL Research University CNRS UMR 3215, INSERM U934, 26 Rue d'Ulm, 75248 Paris Cedex 05 Paris, France. [2]Fish Immunology Laboratory and Center for Research on Pandemic Resilience, Faculty of Life Sciences, Andres Bello University, Santiago 8370146, Chile. [3]Institut Pasteur, Université Paris-Cité, UMR CNRS 6047, Genetics of Biofilms Laboratory, Department of Microbiology, Paris, France. [4]Department of Medicine Solna (MedS), Karolinska Institutet, and Division of Immunology and Respiratory Medicine, Karolinska University Hospital, Solna, and Center for Molecular Medicine, Karolinska University Hospital, SE-171 76 Stockholm, Sweden. [5]Division of Clinical Immunology, Department of Laboratory Medicine (Labmed), Karolinska Institutet, 141 52 Huddinge, Sweden. [6]Host-Microbe Interactomics, Animal Sciences Group, Wageningen University & Research, Wageningen, Netherlands. ✉E-mail: pedro.hernandez-cerda@curie.fr

studies have shown that IL-26 can directly kill bacteria by pore formation in a dose-dependent manner (Meller et al, 2015; Hansen et al, 2021), including both pathogenic and commensal strains. This anti-microbial activity of IL-26 could play a role in regulating microbiota composition in vivo with the loss of this function potentially impairing microbiota-dependent gut epithelial homeostasis, a hypothesis that had not been investigated before this study. Zebrafish possess a single orthologue of the human *IL26* gene (Igawa et al, 2006), along with its two receptor chains (Stein et al, 2007). Therefore, this model provides a suitable system for dissecting the receptor-dependent and -independent functions of IL-26 in vivo, including its role in modulating the microbiota and epithelial homeostasis in early life.

In the human gut, IL-26 is produced by both adaptive and innate lymphocytes, with type 3 innate lymphoid cells (ILC3s) being the primary innate source (Hughes et al, 2009; Cella et al, 2009; Cols et al, 2016; Manel et al, 2008). ILCs were found to populate the human intestine early in embryonic development, prior to 12 weeks post-conception (Liu et al, 2021), thus preceding gut colonization by the microbiota. We previously demonstrated that ILCs are present in the adult zebrafish gut, displaying a cell type diversity resembling that of human ILCs (Hernández et al, 2018). However, whether ILCs are present in the zebrafish gut during early life, whether they respond to the colonizing microbiota by producing IL-26, and the in vivo consequences of this expression on gut microbiota and epithelial homeostasis remain unknown.

In this study, we combined zebrafish genetics, transcriptomics, and microscopy with microbiota profiling, gnotobiotics (engineering microbiota composition), and gut bacterial infection tools to uncover a novel role for IL-26 in regulating gut homeostasis during early life. We report that the loss of IL-26 resulted in increased proliferation in gut epithelial progenitors and elevated DNA damage in absorptive enterocytes in the posterior larval gut, a segment that functionally resembles the human colon (Wei et al, 2023). By characterizing these phenotypes in zebrafish lacking the IL-26 receptor and zebrafish reared germ-free, we observed that IL-26 regulates epithelial cell proliferation and DNA damage independently of its receptor-mediated signaling but in a microbiota-dependent manner. Microbiota profiling, combined with microbiota transfer experiments, revealed that epithelial dysregulation in *il26*-/- larvae results from gut dysbiosis. Notably, we showed that IL-26 bactericidal activity is conserved in zebrafish which may explain dysbiosis in *il26*-/- larvae. Furthermore, IL-26 protected the gut from bacterial infection by regulating bacterial loads, DNA damage, and cytokine expression. Finally, we identified ILCs as the main cell source of IL-26 in the developing larval gut. Overall, our findings suggest a circuit in which early microbial colonization of the developing gut induces IL-26 production in ILCs, which in turn helps establish a healthy microbiota composition, possibly through its intrinsic antibacterial properties, thereby maintaining epithelial homeostasis.

## Results

### IL-26 regulates cell proliferation and DNA damage in the zebrafish larval gut

To study IL-26 in the developing gut, we first analyzed its expression in dissected guts from wild-type (WT) larvae at 3, 5, and 7 days post-fertilization (dpf). Reverse transcriptase PCR (RT-PCR) showed that *il26* is expressed at all three timepoints (Appendix Fig. S1A). Furthermore, RT-qPCR showed no significant differences in *il26* expression levels between these timepoints (Appendix Fig. S1B).

To determine the functions of IL-26, we generated *il26*-deficient zebrafish using the CRISPR/Cas9 system (Appendix Fig. S2A,B). Clutchmate WT and *il26*-/- adult zebrafish were maintained in separate tanks and used to generate WT and *il26*-/- experimental larvae. To characterize the consequences of IL-26 loss on the early-gut transcriptional program, we performed bulk RNA-sequencing (RNA-seq) on dissected guts from 5-dpf *il26*-/- and WT larvae (Fig. 1A). 291 genes were upregulated and 275 downregulated in the guts of *il26*-/- (Dataset EV1). To complement our loss-of-function approach, we performed transcriptomic analysis upon IL-26 overexpression. To this end, recombinant zebrafish IL-26 (rzIl26) or bovine serum albumin (BSA) were injected into the gut and swim bladder of 5-dpf WT larvae, followed by RNA-seq on dissected guts 1 h post-injection (hpi) (Fig. 1A). In this condition, 1759 genes were upregulated and 2022 were downregulated (Dataset EV2).

To infer the affected biological pathways and processes, we performed Kyoto encyclopedia of genes and genomes (KEGG) and gene ontology (GO) analyses on both datasets (Fig. EV1A–D). KEGG pathway analysis on *il26*-/- guts revealed activation of "cell cycle" (Fig. EV1A), with upregulation of genes such as *myca* (Myc proto-oncogene a), a transcription factor that promotes proliferation (Zacarías-Fluck et al, 2024); and *ccnh* (cyclin H), which controls cell cycle progression and promotes cancer growth (Mao et al, 2021; Patel et al, 2016; Peng et al, 2020) (Fig. 1B). In contrast, KEGG pathway analysis upon rzIl26 injection showed suppression of "DNA replication" (Fig. EV1C), with downregulation of genes such as *myca* and *ccnh* (Fig. 1C). Furthermore, GO analysis showed the suppression of "DNA repair" upon rzIl26 injection (Fig. EV1D), with downregulation of several DNA repair genes such as *brca2* (breast cancer gene 2 DNA repair associated (Sadeghi et al, 2020)); *rad51b* (RAD51 paralog B (Wassing and Esashi, 2021)); and *ercc1* (excision repair cross-complementation group 1 (Bohanes et al, 2011)) (Fig. 1D). Collectively, our transcriptomic analysis upon IL-26 loss-of-function and overexpression suggested a potential role for IL-26 in regulating cell proliferation and DNA damage in the gut.

To investigate this possibility, 5-dpf *il26*-/- and WT larvae were incubated with EdU for 3 h, followed by EdU and γH2AX staining to assess proliferation and DNA damage, respectively. *il26*-/- showed higher numbers of EdU-positive cells in the posterior gut. A similar, albeit statistically non-significant, trend was observed in the mid and anterior gut (Fig. EV2A,B). γH2AX-positive cells were mainly detected in the posterior gut of *il26*-/- (Fig. EV2C). This segment of the larval gut is functionally equivalent to the mammalian colon (Wei et al, 2023), the part of the intestine where ulcerative colitis (UC) predominately takes place. Since polymorphisms in the *IL26* gene are a risk factor for UC (Silverberg et al, 2009), we focused our subsequent analyses on the posterior larval gut and found that *il26*-/- larvae showed a higher number of both γH2AX-positive and EdU-positive cells compared to WT controls (Fig. 1E–G). These data indicate an elevated DNA damage response and increased cell proliferation in the posterior larval gut in the absence of IL-26.

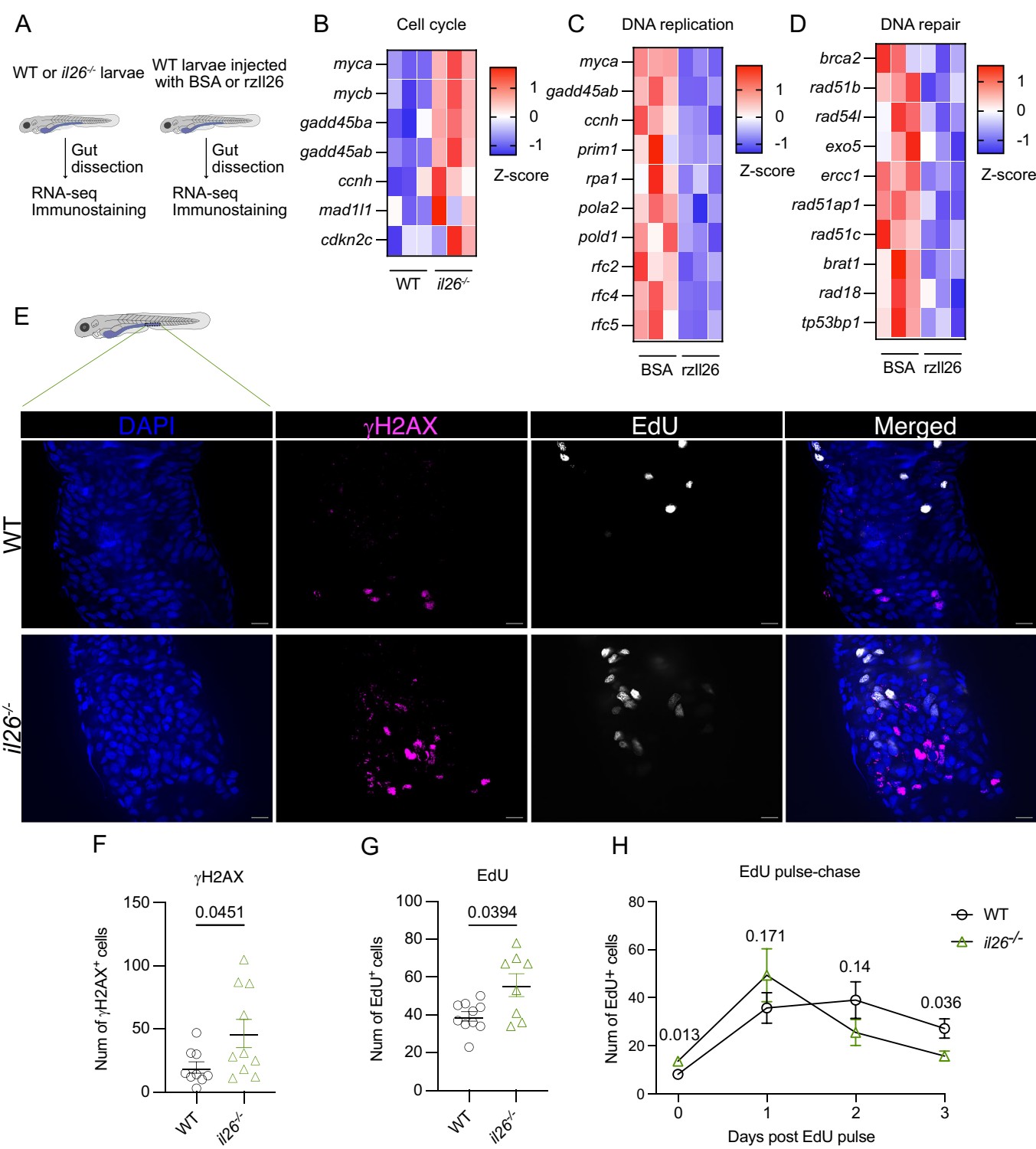

To determine whether IL-26 prevents DNA damage in the gut or induces unresponsiveness to baseline DNA damage, γH2AX staining was performed on 5-dpf WT and *il26⁻/⁻* larval guts 5 h following rzIl26 injection in the gut and swim bladder. rzIl26 administration did not affect the number of γH2AX-positive cells in either WT or *il26⁻/⁻* larvae (Appendix Fig. S3A). This result argues against a rapid transcriptional role for IL-26 in causing

unresponsiveness to DNA damage. Moreover, γH2AX staining in *il26⁻/⁻* appeared pan-nuclear rather than focal (Fig. 1E), a pattern typically associated with widespread DNA damage, as opposed to discrete γH2AX foci associated with enhanced responsiveness to DNA damage. Together, these results support the hypothesis that IL-26 plays a protective role in preventing the accumulation of true DNA lesions in the posterior gut, rather than only modulating the

**Figure 1. Increased proliferation and DNA damage in the gut of *il26-/-* zebrafish larvae.**

(**A**) A schematic of the experimental setup. (**B–D**) Heatmaps of z-scores of genes associated with cell cycle in the loss-of-function dataset (**B**), DNA replication in the overexpression dataset (**C**), and DNA repair in the overexpression dataset (**D**). (**E**) Representative images of EdU and γH2AX staining in WT and *il26-/-* 5-dpf posterior larval guts. Scale bars: 10 µm. (**F, G**) Quantification of γH2AX (**F**) and EdU (**G**) staining in WT and *il26-/-* 5-dpf posterior larval guts. (**H**) EdU pulse-chase analysis in the posterior gut of WT and *il26-/-* larvae. Data information: (**F–H**) Data are presented as mean ± SEM. Sample sizes were as follows ((**F**): $n_{WT} = 9$, $n_{il26-/-} = 10$; (**G**): $n_{WT} = 10$, $n_{il26-/-} = 8$; (**H**): $n_{WT\ 0d} = 21$, $n_{WT\ 1d} = 9$, $n_{WT\ 2d} = 8$, $n_{WT\ 3d} = 16$, $n_{il26-/-\ 0d} = 18$, $n_{il26-/-\ 1d} = 6$, $n_{il26-/-\ 2d} = 9$, $n_{il26-/-\ 3d} = 20$). n represents the number of biological replicates (minimum of 3 independent experiments). Statistical analyses were performed by edgeR package in R (**B–D**) and Mann–Whitney test (**F–H**). Source data are available online for this figure.

DNA damage response. This protective function of IL-26 cannot be recapitulated by a single high-dose administration of the protein.

In addition, rzIl26 injection did not alter the number of EdU-positive cells in either WT or *il26-/-* larvae (Appendix Fig. S3B), suggesting that IL-26-mediated modulation of proliferation is not driven by a rapid transcriptional response. We acknowledge that our current exogenous administration model may not faithfully recapitulate the endogenous expression of IL-26. Differences in concentration, timing, and delivery route may limit its effectiveness. Therefore, the absence of an effect of IL-26 administration on EdU and γH2AX analyses does not conclusively rule out an effect of IL-26 supplementation on these processes. Further experiments with more physiologically relevant delivery may be necessary to fully address this question.

To better understand the physiological consequences of the observed phenotypes in *il26-/-* larvae, epithelial turnover in the posterior gut was quantified by performing EdU pulse-chase in 5-dpf *il26-/- TgBAC(cldn15la-GFP)* larvae, which specifically labels the basolateral membrane of gut epithelial cells (Alvers et al, 2014) (Fig. EV3A). In accordance with the results above, *il26-/-* larvae showed a higher number of EdU-positive cells than WT at 0 day post-EdU pulse. However, at 3 days post-EdU pulse, the mutants had a lower number of EdU-positive cells compared to WT (Fig. 1H), indicating faster epithelial turnover in the *il26-/-* background. Additionally, no differences were observed in gut length (Fig. EV3B,C), or gut barrier integrity (Fig. EV3D,E) between 5-dpf *il26-/-* and WT larvae. Moreover, immune phenotyping based on quantification of gut neutrophil numbers (Fig. EV3F,G) and gene expression of *tnfa* and *il1b* (Fig. EV3H,I) revealed no significant differences between *il26-/-* and WT larvae.

The zebrafish immune system is morphologically and functionally mature by 5 weeks of age (Lam et al, 2004). At this stage, the intestine undergoes key structural and functional changes, including the establishment of intestinal folds, increased complexity of the epithelial architecture, and stabilization of the microbiota (Li et al, 2020; Stephens et al, 2016). To determine whether the increased proliferation and DNA damage observed in *il26-/-* larvae persist through the period of gut and adaptive immune maturation, we quantified EdU and γH2AX staining in 5-week post-fertilization (wpf) juvenile *il26-/-* and WT fish. No differences were observed in either EdU or γH2AX numbers (Fig. EV4A–C). Additionally, we detected no differences in the expression levels of cell cycle markers (*myca, mycb, mych*; Fig. EV4D–F) or DNA repair genes (*brca2, p53*; Fig. EV4G,H). Moreover, expression levels of the cytokines *il1b*, *tnfa*, and *il10*, used to assess inflammation, were also comparable between *il26-/-* and WT juvenile fish. We further followed *il26-/-* fish growth into adulthood (3 months post-fertilization) and found no gross morphological abnormalities (Appendix Fig. S4A,B). These fish survived to adulthood in proportions consistent with

Mendelian genetics (Chi-Square = 1.783, *p*-value = 0.4101, Dataset EV3). Together, our data demonstrate that IL-26 loss leads to increased epithelial cell proliferation, turnover, and DNA damage in the posterior early-life gut, phenotypes that are resolved in later juvenile and adult stages.

## IL-26 suppresses proliferation in epithelial progenitors and DNA damage in posterior absorptive enterocytes

To gain a deeper insight into the phenotypes of increased proliferation and DNA damage in *il26-/-* guts and to pinpoint the affected cell types, we performed single-cell RNA sequencing (scRNA-seq) on whole dissected guts from 5-dpf *il26-/-* and WT larvae. We profiled the gene expression of 13,256 individual cells (7420 WT; 5836 *il26-/-*) (Fig. 2A), with a median of 1517 detected genes per cell. We next applied graph-based clustering on our integrated dataset (Fig. 2B) and identified 24 distinct clusters based on the expression of known markers (Dataset EV4).

In agreement with our cell proliferation observations, the number of epithelial progenitors in *il26-/-* guts was higher than in WT (45 vs. 26 per 1000 detected cells, *p*-value = 0.0319) (Fig. 2C). Moreover, the expression levels of certain cell cycle and stemness markers, such as *stat3* (Peron et al, 2020) and *prmt1* (Tavakoli et al, 2022) were elevated in a few cell populations in *il26-/-* compared to WT, with the most consistent upregulation observed in epithelial progenitors (Fig. 2D). These findings further reinforce the conclusion that IL-26 loss leads to increased proliferation in epithelial progenitors in the larval gut.

Next, to identify the cell types exhibiting higher DNA damage in the guts of *il26-/-*, we analyzed expression levels of several DNA damage-associated genes such as *tp53* and *rad50* across each cell cluster (Fig. 2E). These genes were highly expressed in *il26-/-* compared to WT in several cell populations, including Absorptive_ECs_1, lysosome-rich enterocytes (LREs), and Absorptive_ECs_4. LREs are highly endocytic vacuolated cells that are found in the midgut of zebrafish larvae (Park et al, 2019). Since our γH2AX staining showed a signal only the posterior gut in *il26-/-*, LREs are unlikely to be the cells accumulating DNA damage. Notably, Absorptive_ECs_1 cells were less abundant in *il26-/-* compared to WT (105 vs. 175 per 1000 detected cells, *p*-value < 0.0001), unlike LREs (Fig. 2C). Therefore, our in silico analysis suggests that absorptive enterocytes accumulate DNA damage in *il26-/-*. To validate these findings, γH2AX staining was performed in *il26-/- TgBAC(cldn15la-GFP)* larvae (Fig. 3A). γH2AX-positive cells were GFP-positive, confirming that DNA damage occurs in epithelial cells in the posterior gut of *il26-/-*. To further identify the affected cells, we used the 2F11 antibody which recognizes annexin A4 (*anxa4*) (Zhang et al, 2014) and is commonly used to label secretory cells in zebrafish larvae (Crosnier

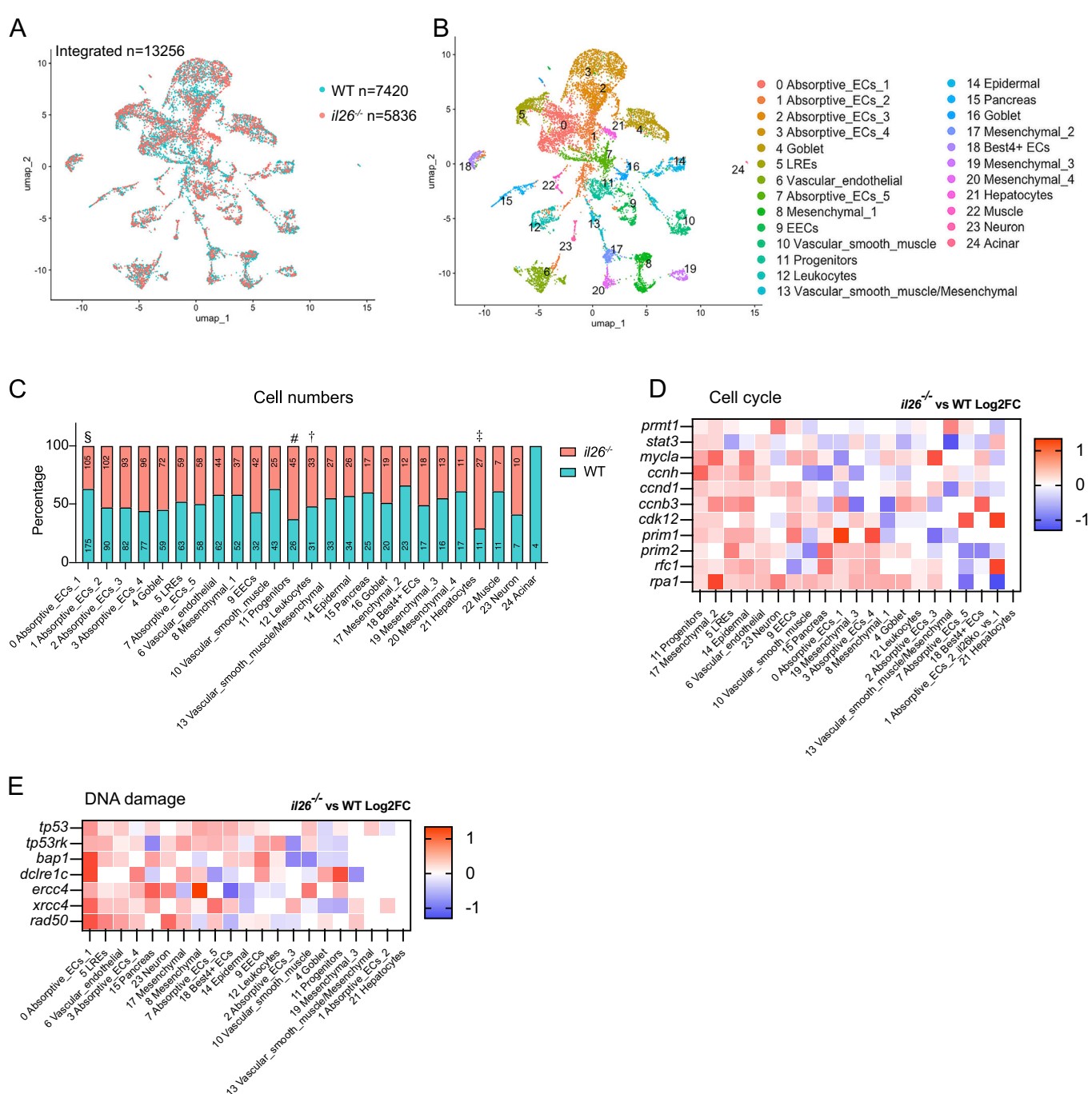

**Figure 2. Single-cell RNA sequencing reveals increased proliferation in epithelial progenitors, elevated DNA damage in absorptive enterocytes, and altered abundance of epithelial cell populations in *il26*⁻/⁻ larval guts.**

(A, B) UMAP Dimensional reduction projection of scRNA-seq dataset of dissected guts of 5-dpf *il26*⁻/⁻ and WT larvae. Cells are colored by genotype (A) or cell type (B). (C) Numbers and percentages of detected cells per cell population, normalized to 1000 detected cells. Statistical significance was assessed using a binomial test. *p*-values: § <0.0001; # 0.0385; † 0.0319; ‡ 0.0139. (D, E) Expression of genes associated with cell cycle (D) or DNA damage (E), colored by log2FC expression in *il26*⁻/⁻ compared to WT across each cell cluster.

et al, 2005). Consistent with this, *anxa4* showed minimal expression in absorptive enterocytes and higher expression in goblet cells, enteroendocrine cells (EECs), Best4+ ECs, and epithelial progenitors (Fig. 3B). Co-staining for γH2AX and 2F11 revealed no colocalization (Fig. 3C), suggesting that γH2AX-

positive cells are absorptive rather than secretory. Overall, our single-cell transcriptomic and in situ analyses suggest that IL-26 suppresses cell proliferation in epithelial progenitors and DNA damage in posterior absorptive enterocytes in the developing larval gut.

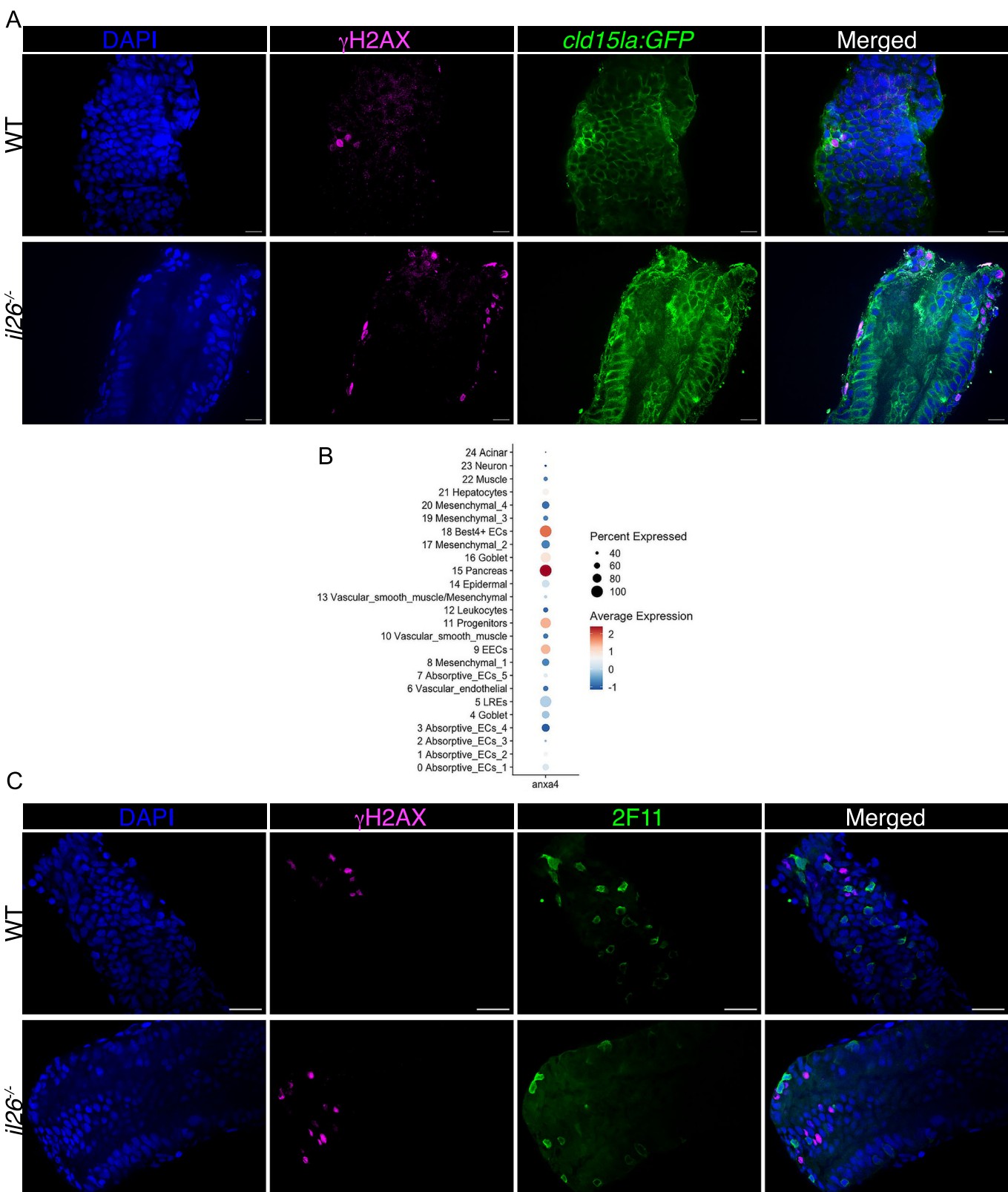

**Figure 3.   IL-26 loss leads to increased DNA damage in absorptive enterocytes in the posterior larval gut.**

(A) γH2AX staining in TgBAC(cldn15la-GFP) in WT and *il26⁻/⁻* genetic backgrounds. Scale bars: 10 μm. (B) Dor plot of *anxa4* expression in the integrated scRNA-seq dataset of WT and *il26⁻/⁻* guts. (C) Co-immunostaining for γH2AX and 2F11in WT and *il26⁻/⁻*. Scale bars: 10 μm. Source data are available online for this figure.

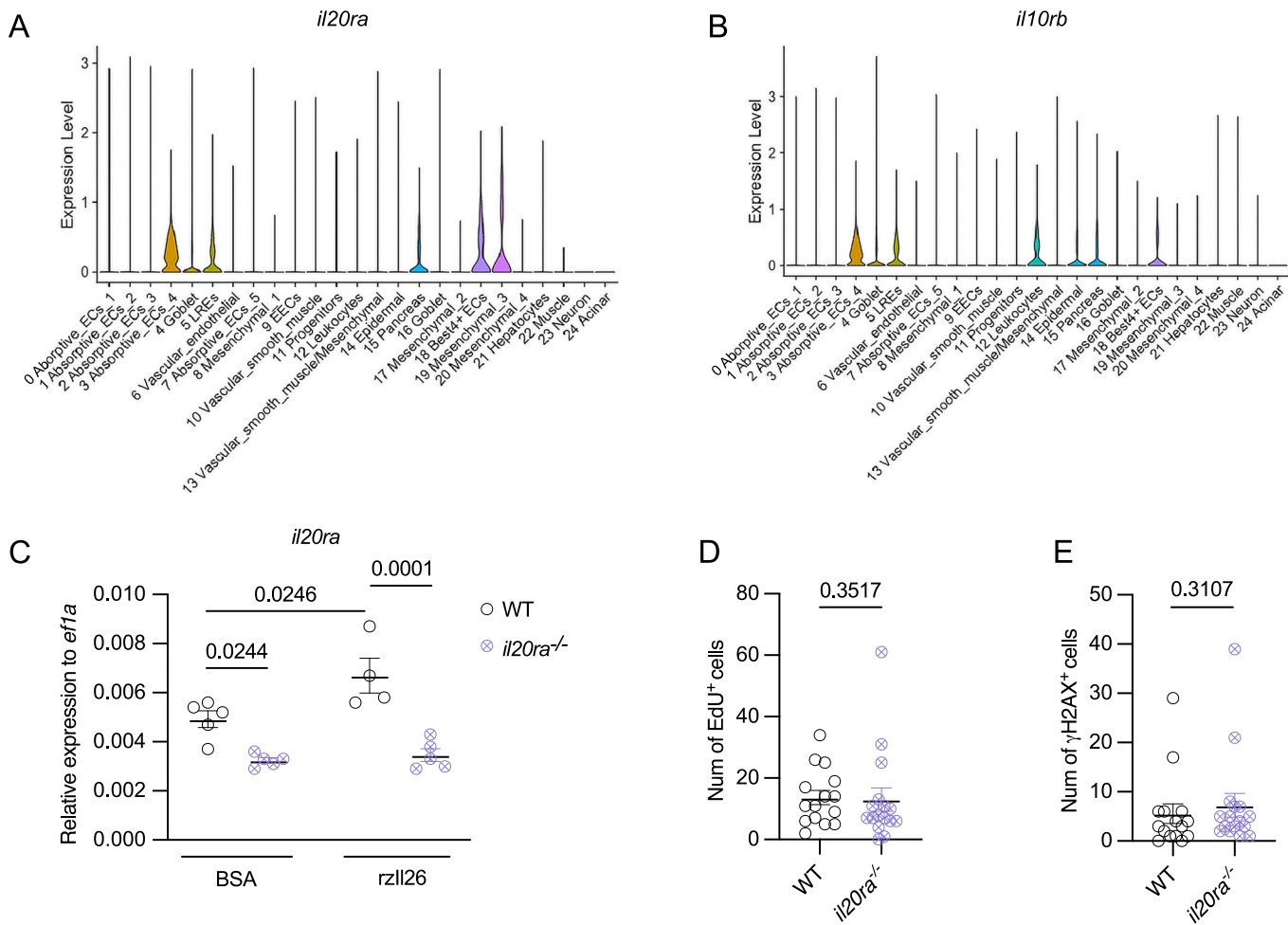

**Figure 4. IL-26 modulation of proliferation and DNA damage in the gut is receptor-independent.**

(A, B) Violin plot of *il20ra* (A) and *il10rb* (B) expression in the integrated scRNA-seq dataset of WT and *il26*-/- guts. (C) qRT-PCR analysis of *il20ra* in dissected guts of 5-dpf WT and *il20ra*-/- larvae 1 h post BSA or rzIl26 administration. (D, E) Quantification of EdU staining (E) and γH2AX staining (F) in WT and *il20ra*-/- 5-dpf posterior larval guts. Data information: (A, B) Violin plots show normalized (log-transformed) gene expression in 7420 WT cells and 5836 *il26*-/- cells. (C–E) Data are presented as mean ± SEM. Sample sizes were as follows ((C): $n_{WT\ BSA} = 5$, $n_{WT\ rzIl26} = 4$, $n_{il26-/-\ BSA} = 5$, $n_{il26-/-\ rzIl26} = 5$; (D): $n_{WT} = 15$, $n_{il26-/-} = 16$; (E): $n_{WT} = 15$, $n_{il26-/-} = 16$). n represents the number of biological replicates (minimum of 3 independent experiments). Statistical significance was determined by two-way ANOVA (C), and Mann–Whitney test (D, E). Source data are available online for this figure.

## IL-26 modulation of epithelial homeostasis in the gut is receptor-independent

The IL-26 receptor complex in humans is composed of two subunits: IL-10RB and IL-20RA (Sheikh et al, 2004; Hör et al, 2004). IL-10RB is shared among IL-10, IL-22, IL-28A, IL-28B, IL-29, and IL-26, whereas IL-20RA is utilized by IL-26, IL-19, IL-20, and IL-24. Notably, the combination of IL-10RB and IL-20RA is unique to IL-26. To elucidate the role of IL-26 receptor signaling in regulating proliferation and DNA damage in gut epithelial cells, we first analyzed the expression profile of these two subunits in our scRNA-seq dataset. Their expression was not detected in either Absoptive_ECs_1 or epithelial progenitors (Fig. 4A,B). However, both were expressed in other cell types in the gut, such as Absoptive_ECs_4 and LREs. To decipher whether IL-26 receptor signaling plays a role in the phenotypes observed in *il26*-/-, we generated *il20ra*-deficient zebrafish using the CRISPR/Cas9 system

(Appendix Fig. S5A,B). To validate this mutant line, we quantified *il20ra* gene expression, as a downstream target gene of IL-26 receptor signaling, at steady state and following rzIl26 administration. At steady state, *il20ra* expression in the gut was significantly reduced in *il20ra*-/- compared to WT (Fig. 4C). Moreover, rzIl26 injection strongly induced *il20ra* expression in WT but failed to do so in *il20ra*-/- larval guts (Fig. 4C), indicating effective disruption of IL-26 receptor-mediated signaling. Interestingly, *il20ra*-/- larvae showed similar levels of EdU and γH2AX compared to WT (Fig. 4D,E; Appendix Fig. S5C). These findings support the hypothesis that IL-26 regulates epithelial cell proliferation and DNA damage in the early-life gut in a receptor-independent manner.

## Analysis of IL-26 receptor-independent functions

Human IL-26 protein has receptor-independent functions. For example, human IL-26 protein (hIL26) can form complexes with

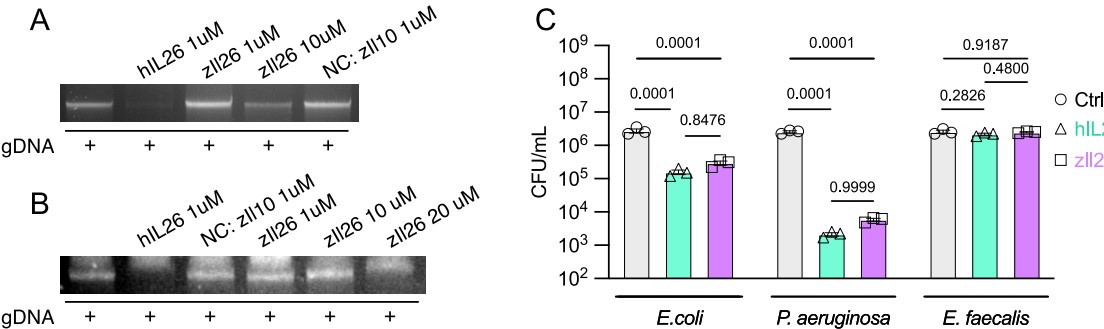

**Figure 5. Analysis of IL-26 receptor-independent functions.**

(A, B) Gel migration assay of genomic DNA incubated with several cytokines. (C) Quantification of colony-forming units of different bacterial species after incubation with human or zebrafish IL-26 proteins at a concentration of 8 µM. Error bars show means ± SEM. Data information: (C) Data are presented as mean ± SEM. Sample sizes were as follows ((C): n = 3). n represents the number of biological replicates (minimum of 3 independent experiments). Statistical significance was determined by two-way ANOVA (C). Source data are available online for this figure.

DNA and activate intracellular TLR9 in the absence of the IL-26 receptor (Meller et al, 2015). Since TLR9 regulates cell proliferation (Jiang et al, 2024; Parroche et al, 2016) and DNA damage (Jovasevic et al, 2024), we investigated whether rzIl26 can form DNA complexes using gel migration assays. In contrast to hIL26, incubating rzIl26 with genomic DNA (gDNA) at a concentration of 1 µM did not block DNA migration (Fig. 5A). At 10 µM, rzIL26 only partially impeded DNA migration (Fig. 5A), while at 20 µM, a clear inhibition was observed (Fig. 5B). These results reveal a weaker DNA-binding capacity of zebrafish IL-26 compared to the human counterpart.

In addition, hIL26 exhibits intrinsic bactericidal activity, having been shown to kill *E. coli* and *P. aeruginosa* but not *E. faecalis*, at a concentration of 8 µM (Meller et al, 2015). We compared the bactericidal activity of human and zebrafish IL-26 proteins by incubating each protein at a concentration of 8 µM with these bacteria, followed by colony-forming units (CFU) analysis. Both proteins killed *E. coli* and *P. aeruginosa* to similar levels but neither protein killed *E. faecalis* (Fig. 5C). These results show that the intrinsic bactericidal activity of IL-26 is conserved in zebrafish, with similar specificity and efficacy.

## IL-26 maintains gut epithelial homeostasis by controlling the composition of the microbiota

Having found that IL-26 regulates gut epithelial cell proliferation and DNA damage in a receptor-independent manner and that zebrafish IL-26 protein has bactericidal properties, we wondered whether the lack of this function in *il26*-/- might result in an altered microbiota composition, leading to increased proliferation and DNA damage. To uncover whether IL-26 loss leads to dysbiosis in zebrafish larval guts, we profiled the composition of the microbiota using 16S rRNA-seq on dissected guts from 5-dpf WT and *il26*-/- larvae. Redundancy analysis (RDA) of the detected bacterial families revealed clear separation between WT and *il26*-/- (p-value = 0.002, permutation test; Fig. 6A; Dataset EV5), pointing to differences in their gut bacterial communities. Indeed, several bacterial families were differentially represented in the WT

compared to *il26*-/- gut (Fig. 6B,C). Notably, *Enterobacteriaceae*, correlated with IBD in humans (Khorsand et al, 2022), were more abundant in *il26*-/- larval guts (Fig. 6C). These data demonstrate that IL-26 shapes the composition of the gut microbiota in the zebrafish larval gut.

To determine the role of the microbiota in the observed phenotypes, we first performed bulk RNA-seq analysis on dissected guts from 5-dpf WT and *il26*-/- larvae reared under germ-free (GF) *il26*-/- conditions. 319 genes were upregulated and 307 down-regulated in the guts of *il26*-/- GF larvae (Dataset EV6). Notably, genes related to cell cycle and DNA repair were not differentially expressed in *il26*-/- GF compared to WT GF larvae (Fig. 6D), suggesting that the increased proliferation and DNA damage in *il26*-/- under conventional (CV) conditions are absent in GF conditions. To validate this, we reared WT and *il26*-/- larvae under CV or GF conditions and quantified gut cell proliferation and DNA damage. EdU- and γH2AX-positive cells were less abundant in *il26*-/- GF compared to *il26*-/- CV (Fig. 6E,F; Appendix Fig. S6A,B). Furthermore, *il26*-/- GF showed similar levels of proliferation and DNA damage as WT GF. These data indicate that IL-26 suppresses cell proliferation and DNA damage in the zebrafish gut in a microbiota-dependent manner.

Next, we hypothesized that if the altered microbiota in *il26*-/- larvae was driving the observed elevated proliferation and DNA damage, then introducing WT microbiota to *il26*-/- larvae might rescue these phenotypes. To verify this, *il26*-/- larvae were maintained GF until 3 dpf, then co-housed with WT CV larvae by transferring them into WT CV plates, separated by a 40 µm sterile strainer to allow microbial exchange without direct contact, until 5 dpf. Co-housed *il26*-/- larvae displayed cell proliferation and DNA damage levels similar to those in WT CV (Fig. 6E,F; Appendix Fig. S6A,B), suggesting that WT microbiota mitigated the increased proliferation and DNA damage in *il26*-/- larvae. Notably, co-housing WT GF with *il26*-/- CV did not affect EdU or γH2AX levels in co-housed WT guts, indicating that WT larvae are resilient to microbial dysbiosis associated with IL-26 deficiency. Altogether, these observations suggest that *il26*-deficiency leads to dysbiosis, consequently resulting in impaired epithelial homeostasis.

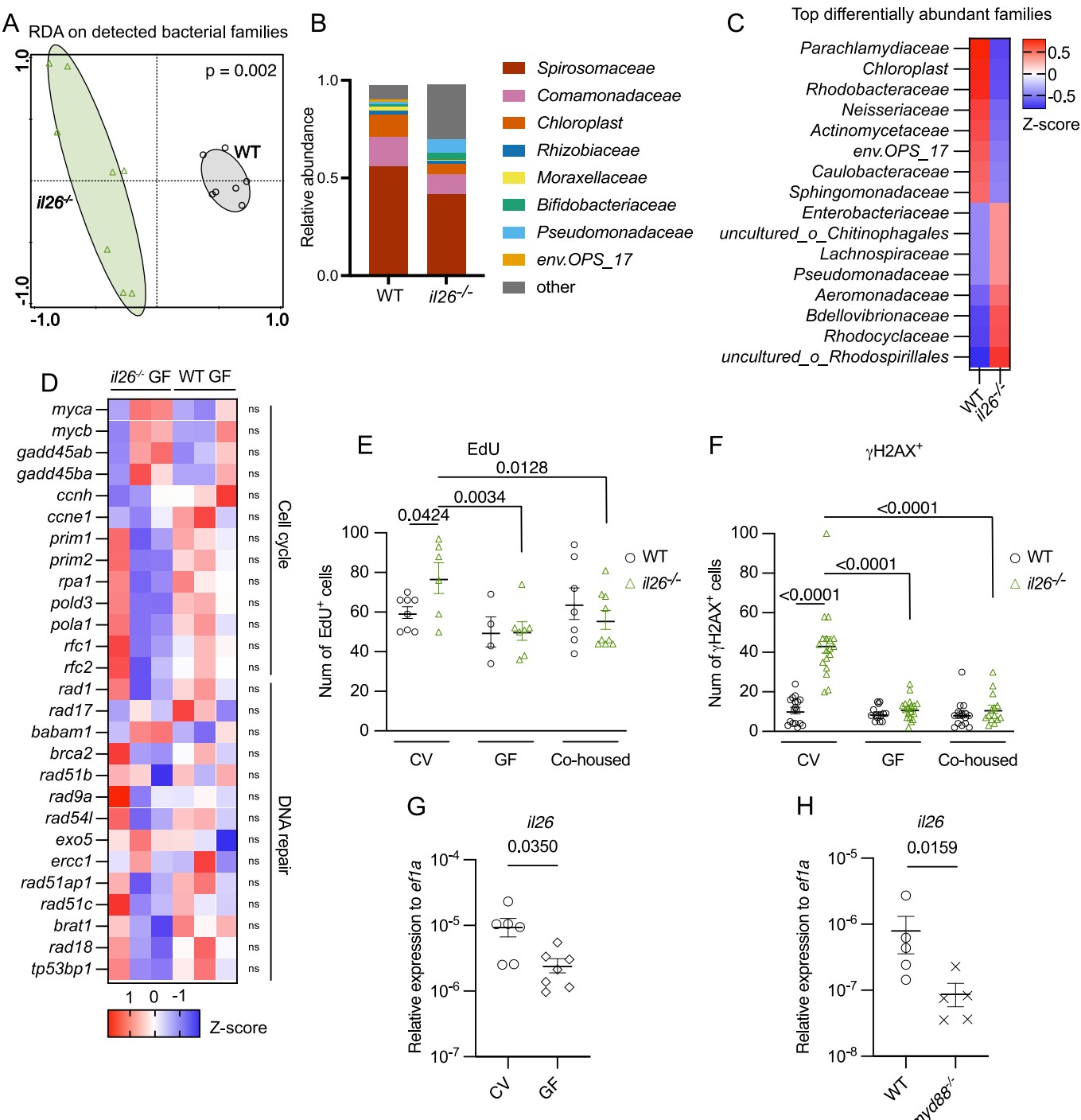

**Figure 6. Interactions between IL-26 and the microbiota regulate epithelial proliferation and DNA damage in the larval gut.**

(A) RDA plot on the bacterial families detected by 16S rRNA-sequencing on 5-dpf WT and *il26$^{-/-}$* larval intestines. (B) Relative abundance of the detected bacterial families in WT and *il26$^{-/-}$* larval guts. Only families with an abundance greater than 1% in WT are shown. (C) Heatmap of z-scores of the relative abundance of the top differentially abundant families in WT and *il26$^{-/-}$* larval guts. (D) Heatmap of z-scores of genes associated with cell cycle and DNA repair in *il26$^{-/-}$* GF compared to WT GF. (E, F) Quantification of EdU staining (E) and γH2AX staining (F) in WT and *il26$^{-/-}$* larval guts reared CV, GF, or co-housed. (G) qRT-PCR analysis of *il26* in dissected guts of 5-dpf WT larvae reared CV or GF. (H) qRT-PCR analysis of *il26* in dissected guts of WT or *myd88$^{-/-}$* 5-dpf larvae. Data information: (E–H) Data are presented as mean ± SEM. Sample sizes were as follows ((E): $n_{WT\ CV}$ = 8, $n_{WT\ GF}$ = 4, $n_{WT\ Co-housed}$ = 7, $n_{il26-/-\ CV}$ = 6, $n_{il26-/-\ GF}$ = 7, $n_{il26-/-\ Co-housed}$ = 9; (F): $n_{WT\ CV}$ = 17, $n_{WT\ GF}$ = 14, $n_{WT\ Co-housed}$ = 16, $n_{il26-/-\ CV}$ = 18, $n_{il26-/-\ GF}$ = 18, $n_{il26-/-\ Co-housed}$ = 14; (G): $n_{CV}$ = 6, $n_{GF}$ = 7; (H): $n_{WT}$ = 5, $n_{myd88-/-}$ = 5). n represents the number of biological replicates (minimum of 3 independent experiments). Statistical significance was determined by two-way ANOVA (E, F), and Mann–Whitney test (G, H). Source data are available online for this figure.

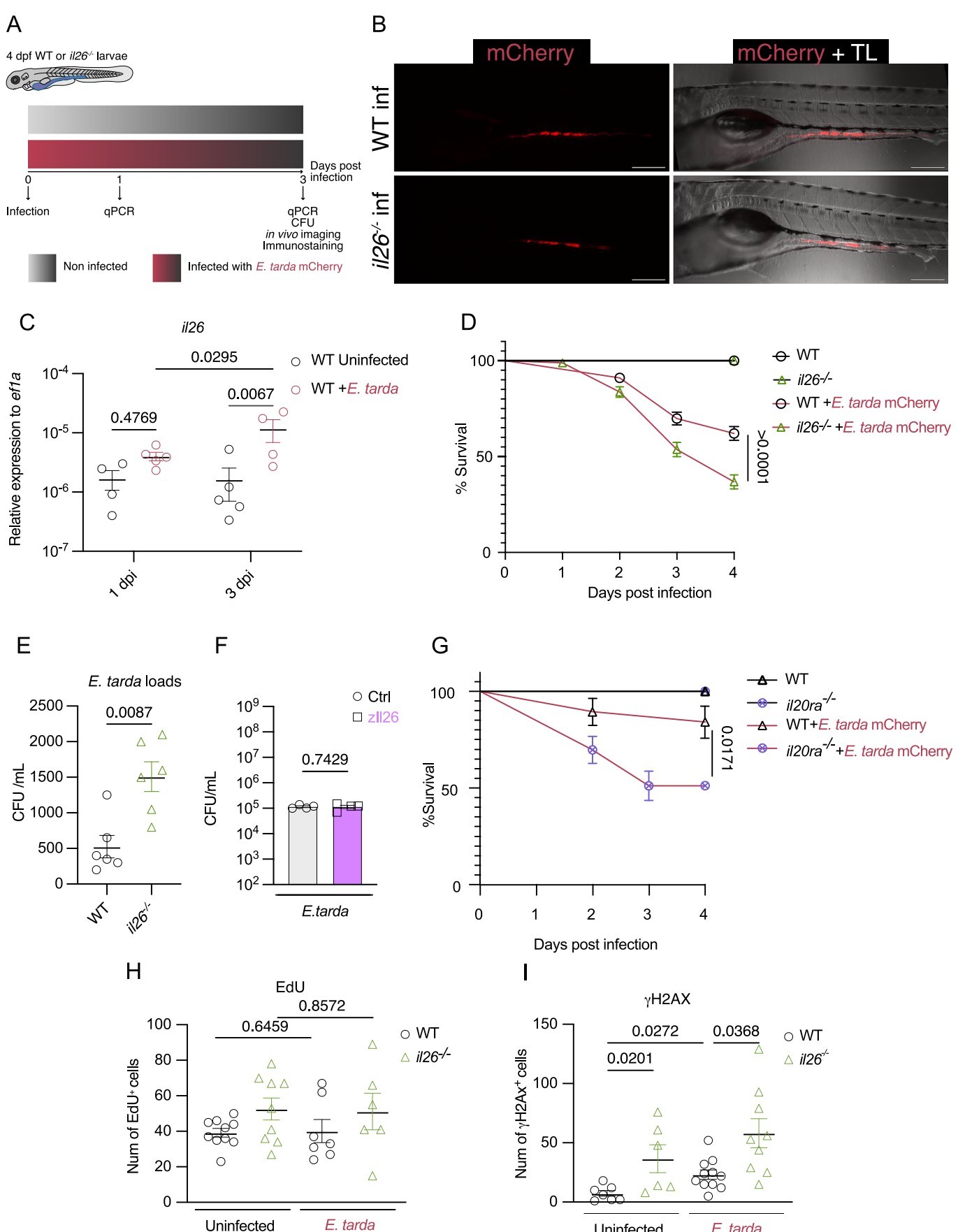

**Figure 7.  IL-26 protects the gut from *E. tarda* bacterial infection.**

(A) Schematic representation of *E. tarda* infection and subsequent analyses. (B) WT and *il26*[-/-] larvae infected with mCherry-labeled *E. tarda* at 3 dpi. Scale bars: 200 μm. (C) qRT-PCR analysis of *il26* in dissected guts of WT larvae at 1 and 3 dpi. (D) Survival analysis of WT and *il26*[-/-] larvae infected with *E. tarda*. (E) Quantification of colony-forming units of *E. tarda* in dissected guts of WT and *il26*[-/-] larvae at 3 dpi. (F) Quantification of colony-forming units of *E. tarda* after incubation zebrafish IL-26 protein at a concentration of 8 μM. (G) Survival analysis of WT and *il20ra*[-/-] larvae infected with *E. tarda*. (H, I) Quantification of EdU (H) and γH2AX (I) staining in WT and *il26*[-/-] larval guts at 3 dpi. Data information: (C, E, F, H, I) Data are presented as mean ± SEM. Sample sizes were as follows ((C): 4–5; (D): $n_{WT\ uninfected} = 164$, $n_{WT\ infected} = 189$, $n_{il26-/-\ uninfected} = 164$, $n_{il26-/-\ infected} = 177$; (E): $n_{WT} = 6$, $n_{il26-/-} = 6$; (F): $n_{ctrl} = 4$, $n_{zIl26} = 4$; (G): $n_{WT\ uninfected} = 12$, $n_{WT\ infected} = 19$, $n_{il20ra-/-\ uninfected} = 12$, $n_{il20ra-/-\ infected} = 43$; (H): $n_{WT\ uninfected} = 10$, $n_{WT\ infected} = 7$, $n_{il26-/-\ uninfected} = 9$, $n_{il26-/-\ infected} = 6$; (I): $n_{WT\ uninfected} = 7$, $n_{WT\ infected} = 11$, $n_{il26-/-\ uninfected} = 7$, $n_{il26-/-\ infected} = 9$). n represents the number of biological replicates (minimum of 3 independent experiments). Statistical significance was determined by two-way ANOVA (C), Gehan-Breslow-Wilcoxon test (D, G), Mann–Whitney test (E, F), and Kruskal-Wallis test (H, I). Source data are available online for this figure.

## Innate sensing of the microbiota induces IL-26 expression in the larval gut

Microbial gut colonization is known to induce transcriptional changes, including cytokine expression (Fu et al, 2021; Koch et al, 2018). To determine whether the microbiota influences *il26* expression in the larval gut, mRNA levels were quantified in 5-dpf CV and GF larval guts. GF larval guts showed lower *il26* expression levels compared to WT (Fig. 6G), indicating that the microbiota induces baseline *il26* expression. To uncover the signaling pathways involved in this process, we measured *il26* in larvae deficient for Myd88, a key adaptor protein in TLR signaling (Kawai et al, 2024). *il26* expression was lower in *myd88*[-/-] larval guts compared to WT controls (Fig. 6H). These results suggest that microbial colonization in the gut induces baseline *il26* expression in the zebrafish larval gut through TLR-dependent innate immune sensing.

## IL-26 protects the gut from bacterial infection

Given that bacterial infections in the gut are known to cause DNA damage and contribute to IBD and intestinal cancer (Yang et al, 2017; Pleguezuelos-Manzano et al, 2020; Prorok-Hamon et al, 2014), we examined whether IL-26 plays a role during gut bacterial infections. We infected WT and *il26*[-/-] larvae via water bath immersion with mCherry-labeled *Edwardsiella tarda*, a Gram-negative bacterium that infects both zebrafish and human guts and is known to cause mortality in zebrafish larvae (Bravo-Tello et al, 2017; Ye et al, 2021; Michael and Abbott, 1993) (Fig. 7A). Live imaging revealed that *E. tarda* accumulated primarily in the mid and posterior gut at 3 days post infection (dpi) in both WT and *il26*[-/-] larvae (Fig. 7B). To know whether this infection model induces *il26* expression, we quantified mRNA levels in dissected WT larval guts at 1 and 3 dpi. We found that *il26* expression was higher in 3-dpi larvae compared to uninfected controls (Fig. 7C). Next, we monitored the survival of *E. tarda*-infected WT and *il26*[-/-] larvae. *il26*[-/-] larvae exhibited higher mortality compared to WT upon infection (Fig. 7D). Furthermore, CFU analysis revealed higher bacterial loads in *il26*[-/-] guts at 3 dpi (Fig. 7E). Together, our data demonstrate that IL-26 contributes to host defense and protects against *E. tarda* infection.

Given that zebrafish IL-26 possesses intrinsic bactericidal activity, we investigated whether its protective effect against *E. tarda* is mediated through this property. To test this, *E. tarda* was incubated with rzIl26, followed by CFU analysis (Fig. 7F). rzIl26 did not reduce bacterial viability, indicating that IL-26 protective function is not due to direct killing of *E.tarda*. This suggested that

IL-26 protectes againt *E.tarda* through its receptor-mediated signaling. To test this hypothesis, we infected *il20ra*[-/-] larvae and monitored survival. Similar to *il26*[-/-] larvae, *il20ra*[-/-] mutants displayed increased susceptibility to *E. tarda* (Fig. 7G). These findings indicate that IL-26 exerts its protective effects upon *E. tarda* infection in a receptor-dependent manner.

Next, we investigated the role of IL-26 in regulating gut epithelial proliferation and DNA damage upon *E. tarda* infection by staining WT and *il26*[-/-] larvae with EdU and γH2AX at 3 dpi. *E. tarda*-infected WT and *il26*[-/-] guts showed no differences in the number of EdU-positive cells compared to their respective uninfected controls (Fig. 7H; Appendix Fig. S7). Interestingly, *E. tarda* infection led to an increased number of γH2AX-positive cells in WT guts (Fig. 7I; Appendix Fig. S7), indicating that *E. tarda* infection induces DNA damage in the posterior larval gut. Moreover, infected *il26*[-/-] guts exhibited greater DNA damage compared to infected WT guts. Our findings indicate that the loss of IL-26 renders gut epithelial cells more susceptible to DNA damage but does not affect their proliferation upon *E. tarda* infection.

Finally, to characterize the immune response in the gut of *il26*[-/-] upon infection, we measured the expression levels of several cytokines in WT and *il26*[-/-] larval guts at 3 dpi. *E. tarda* infection induced the upregulation of *il1b*, *il22*, *tnfa*, and *il10* in WT larvae (Fig. EV5A–D). Similarly, *il26*[-/-] guts showed higher cytokine expression levels upon infection compared to uninfected *il26*[-/-] controls. However, these cytokines showed significantly lower expression levels in infected *il26*[-/-] compared to infected WT larvae.

In sum, we show that *il26*[-/-] larvae are more susceptible to gut bacterial infection, exhibiting higher bacterial loads, increased DNA damage, and an impaired immune response. These findings underscore the critical role of IL-26 and IL-26 receptor-mediated signaling in maintaining gut integrity and regulating immune responses during bacterial infections.

## Innate lymphoid cells are the primary source of IL-26 in the larval gut

Having observed that IL-26-interactions with the microbiota maintain zebrafish gut epithelial homeostasis, we aimed to identify its cellular sources. We first examined *il26* expression in our scRNA-seq dataset, however, it was not detected. This is likely due to the low expression levels of *il26* at steady state as well as the low sequencing depth of current 10X genomics technologies. Since *il26* expression was induced upon inflammation (Fig. 7C), we re-analyzed a published scRNA-seq dataset from guts of larvae incubated with dextran sulfate sodium (DSS), a chemical that

induces gut inflammation (Nayar et al, 2021; Data ref: Nayar et al, 2021). *il26* was mainly expressed in a population of lymphocytes characterized by the expression of novel immune-type receptor 4a gene (*nitr4a*) a specific marker of zebrafish ILCs (Hernández et al, 2018) (Fig. 8A,B). Among ILCs, IL-26 is primarily expressed by ILC3s (Cella et al, 2009; Hernández et al, 2018). To further characterize the identity of *il26*-expressing cells, we analyzed the expression of canonical ILC markers. While a small proportion of ILCs expressed ILC1 markers (*eomesa*, *tnfa*) and ILC2 markers (*gata3*, *il13*), a larger fraction expressed the ILC3 marker *rorc* (Fig. 8B). In addition, *il26*⁺ ILCs predominantly expressed *rorc*, while ILC1 and ILC2 markers were mostly undetected in these cells (Fig. 8C). These findings indicate that ILC3s are the main source of *il26* during inflammation in the larval gut.

To determine the contribution of ILCs to baseline *il26* expression in the zebrafish larval gut at steady state, complementary approaches were employed. First, we measured *il26* in dissected guts of 5-dpf *il2rga⁻/⁻prkdc⁻/⁻* larvae, devoid of adaptive lymphocytes and ILCs (Yan et al, 2019). *il26* levels were lower in *il2rga⁻/⁻prkdc⁻/⁻* compared to *il2rga⁺/⁺prkdc⁻/⁻* (Fig. 8D), indicating that lymphocytes are required for *il26* expression in the larval gut. Second, we utilized *rag1⁻/⁻* larvae, which lack adaptive lymphocytes but still possess ILCs (Petrie-Hanson et al, 2009). *il26* levels in *rag1⁻/⁻* were similar to those in WT larvae (Fig. 8E), indicating that adaptive lymphocytes are not required for *il26* expression. Finally, to visualize *il26* expression in situ in gut ILCs, we performed RNA-FISH for *il26* and *nitr9*, a specific marker of zebrafish ILCs (Hernández et al, 2018). We detected *il26* expression in *nitr9*-positive cells (Fig. 8F). Together, our data indicate that *il26*-expressing ILCs are present in the developing zebrafish gut and are required for *il26* expression as early as 5 dpf.

## Discussion

In this study, we created the first in vivo animal model to study the impact of IL-26 loss-of-function on gut homeostasis. We report that early microbial gut colonization in zebrafish induces IL-26 production in ILCs, which shapes the composition of the microbiota and, in turn, helps maintain epithelial homeostasis (Fig. 9). Our analyses suggest that this function of IL-26 is independent of its canonical receptor complex, composed of IL10RB and IL20RA. In addition, we revealed that zebrafish IL-26 possesses intrinsic antibacterial properties, suggesting that the modulation of gut microbiota composition by IL-26 in early life is mediated through this property (Fig. 9).

The dysregulation of epithelial homeostasis upon IL-26 loss was marked by heightened cell proliferation in gut epithelial progenitors and elevated DNA damage in posterior absorptive enterocytes. Notably, our scRNA-seq analysis showed increased numbers of epithelial progenitors and decreased numbers of Absorptive_ECs_1 in *il26⁻/⁻* (Fig. 2C). These observations underscore how host-microbiota interaction mediated by IL-26 differentially regulates distinct epithelial cell types, paving the way for innovative strategies for targeting the gut epithelium in a cell-specific manner.

The consequences of IL-26 loss on proliferation and DNA damage were not observed at later developmental stages in juvenile fish. This could be attributed to the maturation of the immune system in juvenile zebrafish (Lam et al, 2004) or to feeding starting

at 6 dpf in the animal facility. Profiling of the microbiota in juvenile and adult *il26⁻/⁻* fish could clarify whether the dysbiosis observed in larval stages is resolved, potentially through adaptive immunity, or whether it persists but is mitigated by compensatory mechanisms that limit its impact on the gut epithelium.

Human IL26 is an amphipathic 171-amino acid protein with 30 positively charged residues and an isoelectric point of 10.7. This cationic amphipathic nature is similar to that of classical antimicrobial peptides (AMPs). These properties allow human IL26 to bind to and penetrate bacterial membranes, which are typically negatively charged, thereby killing these bacteria by pore formation in a receptor-independent manner (Meller et al, 2015). It remains unclear which bacterial species are most susceptible to IL26. It is likely that the net surface charge of bacterial membranes plays a role in susceptibility, but this has not yet been systematically studied. In addition, human IL26 has been shown to bind to DNA. However, whether this ability is due to electrostatic interactions between negatively charged DNA and the positively charged IL26 is unclear.

Zebrafish Il26 is an amphipathic 169-amino acid protein with 34 positively charged residues and an isoelectric point of 9.51. The shared physiochemical properties between the human and zebrafish IL-26 proteins prompted us to examine the conservation of IL-26 receptor-independent functions in zebrafish. We demonstrated that zebrafish Il26 killed bacteria to similar levels as the human protein, indicating that the intrinsic bactericidal activity of human IL26 is conserved in zebrafish. Notably, despite the conserved cationic nature of IL-26 between humans and zebrafish, we found that zebrafish Il26 binds DNA with lower affinity than the human protein. Further analyses of the specific structural and physio-chemical properties of human and zebrafish IL-26 proteins may identify the different domains of IL-26 that are involved in receptor binding, DNA binding, and bactericidal activity. This could pave the way for the development of IL-26-derived peptides as novel antibacterial agents, an urgent necessity given the increasing spread of antibiotic-resistant bacteria.

Given that the antimicrobial activity of IL-26 is conserved in zebrafish and that IL-26 regulates proliferation and DNA damage in a receptor-independent manner, we hypothesized that IL-26 suppresses proliferation and DNA damage in gut epithelial cells through modulating the microbiota composition. Our 16S rRNA-seq analysis demonstrated that IL-26 deficiency leads to dysbiosis in the zebrafish larval gut. In addition, co-housing-mediated microbiota transfer from WT to *il26⁻/⁻* restored the high proliferation and DNA damage in *il26⁻/⁻* to WT CV levels. These results support the hypothesis that an altered microbiota composition in *il26⁻/⁻* is responsible for the elevated proliferation and DNA damage. Overall, these data suggest that the loss of IL-26 antimicrobial activity leads to dysbiosis in the gut, which in turn results in higher proliferation and DNA damage in gut epithelial cells.

Interestingly, microbiota transfer from *il26⁻/⁻* to WT did not result in increased proliferation and DNA damage in WT. This suggests that the microbial agents responsible for the phenotypes in *il26⁻/⁻* are not able to colonize WT gut, possibly due to the expression of IL-26 in WT. Microbiota profiling in this condition could help identify the bacterial species that are failing to colonize WT larvae, indicating a potential role for these species in inducing cell proliferation and heightened DNA damage in *il26⁻/⁻* gut epithelial cells.

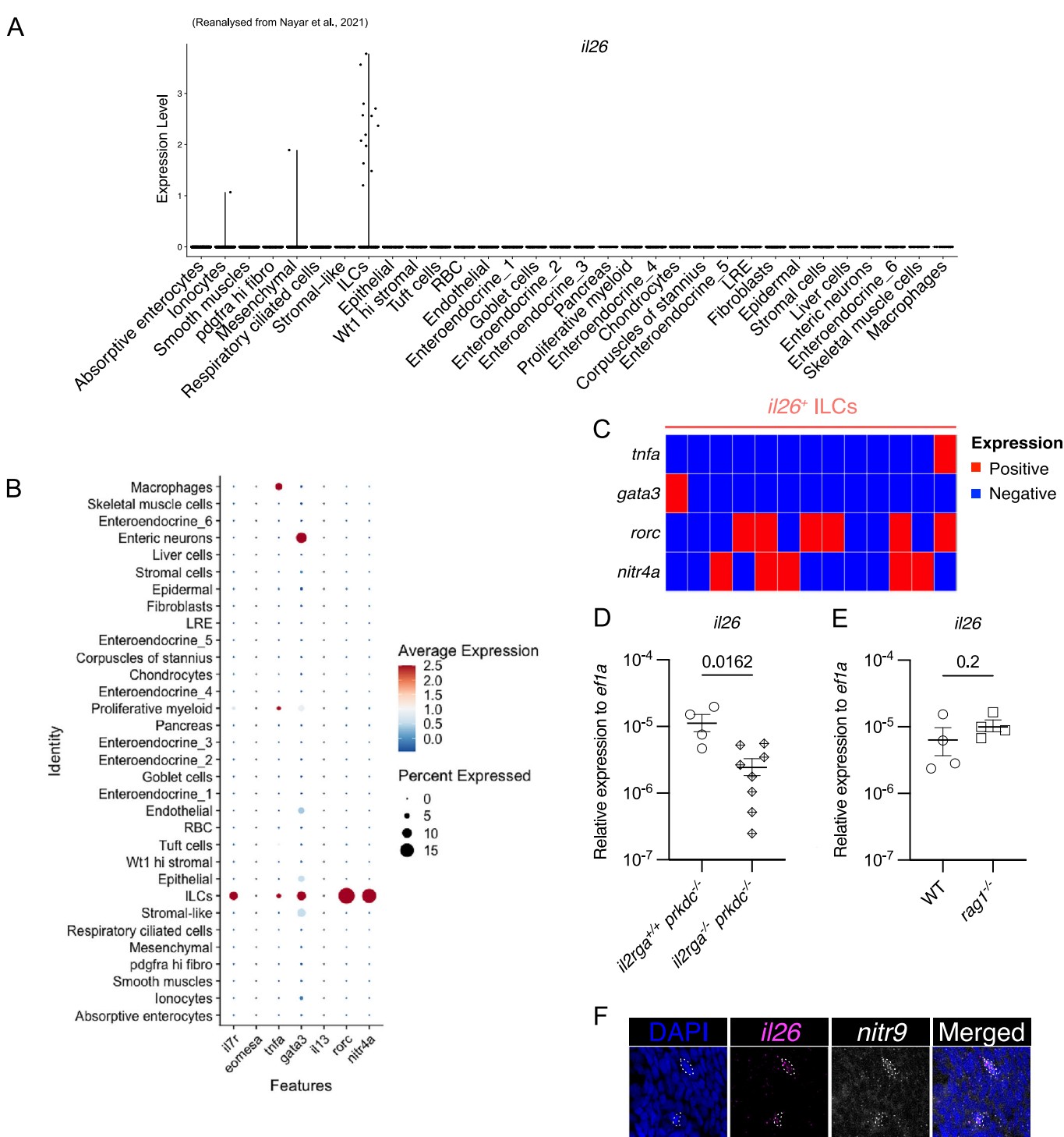

**Figure 8. IL-26 is expressed by ILC3 in the zebrafish larval gut.**

(A) Violin plot of *il26* expression, reanalyzed from (Nayar et al, 2021). (B) Dot plot of ILC markers, reanalyzed from (Nayar et al, 2021). (C) Binary heatmap showing expression of selected genes in individual *il26*⁺ ILCs. (D) qRT-PCR analysis of *il26* in dissected guts of *il2rga*⁺/⁺*prkdc*⁻/⁻ and *il2rga*⁻/⁻*prkdc*⁻/⁻ 5-dpf larvae. (E) qRT-PCR analysis of *il26* in dissected guts of WT or *rag1*⁻/⁻ 5-dpf larvae. (F) RNA-FISH of *il26* and *nitr9* on dissected guts of 5-dpf larvae. Scale bars: 10 μm. Data information: (D, E) Data are presented as mean ± SEM. Sample sizes were as follows ((D): $n_{il2rga+/+} = 4$, $n_{il2rga-/-} = 8$; (E): $n_{WT} = 4$, $n_{rag1-/-} = 4$). n represents the number of biological replicates (minimum of 3 independent experiments). Statistical significance was determined by Mann–Whitney test (D, E). Source data are available online for this figure.

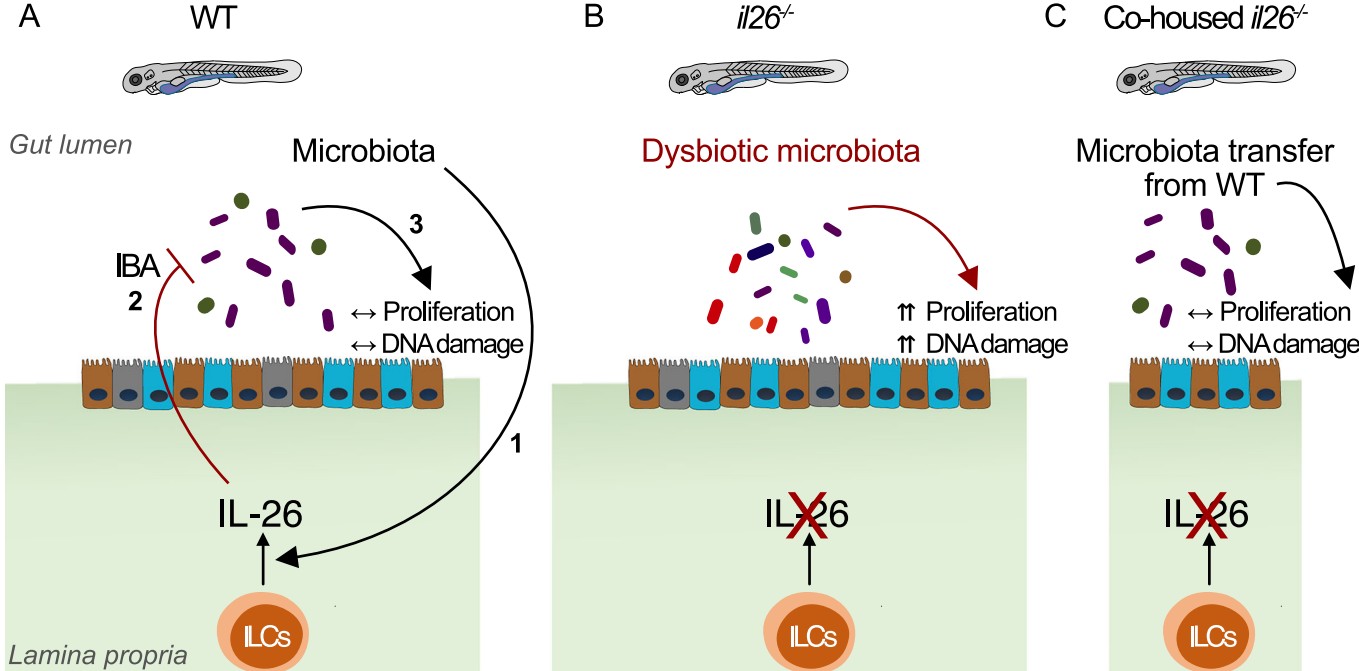

**Figure 9. Working model.**

(A) in the developing gut of WT larvae, microbial colonization induces IL-26 expression in ILCs (1), which in turn regulates gut microbiota composition possibly through its intrinsic bactericidal activity (IBA) (2), thereby maintaining baseline levels of proliferation and DNA damage in gut epithelial cells (3). (B) In *il26⁻/⁻* larvae, the loss of IL-26 IBA leads to dysbiosis, which in turn increases proliferation and DNA damage in the posterior gut epithelium. (C) Co-housing mediated microbiota transfer from WT to *il26⁻/⁻* larvae rescues the high proliferation and DNA damage in *il26⁻/⁻* gut epithelial cells.

Recombinant zebrafish IL-26 administration did not lower proliferation and DNA damage levels in either WT or *il26⁻/⁻* larvae. This finding argues against a direct and rapid transcriptional role for IL-26 in modulating these phenotypes and therefore supports the hypothesis that these phenotypes are caused by dysbiosis. This is because a single exogenous administration of rzIl26 at 5 dpf may not be sufficient to influence microbiota composition, compared to constitutive endogenous expression starting from 3 dpf, when microbial colonization begins. Further studies aiming to profile microbiota composition and deliver IL-26 using more physiologically relevant methods may be necessary to fully evaluate this possibility.

Our 16S rRNA-seq analysis revealed that IL-26 deficiency leads to an increased abundance of *Enterobacteriaceae*, a family of bacteria that includes well-known genera such as *Escherichia coli*. This bacterial family is known to colonize the human colon and is consistently enriched in IBD patients (Baldelli et al, 2021). Moreover, members of the *Enterobacteriaceae* family, such as *pks + Escherichia coli*, have been reported to produce colibactin, a genotoxin capable of alkylating DNA and inducing double-strand breaks in gut epithelial cells (Arthur et al, 2012). Interestingly, we demonstrated that zebrafish IL-26 can kill *E. coli* (Fig. 3D). However, whether the observed increase in DNA damage in the posterior gut of *il26⁻/⁻* is directly attributable to an enrichment of DNA-damage inducing members of *Enterobacteriaceae* and whether this enrichment is a consequence of the loss of IL-26 antibacterial function warrants further investigation. This may inform the development of IL-26 supplementation as a potential therapeutic strategy for IBD patients with elevated levels of *Enterobacteriaceae*.

IBD patients carrying the *IL26* risk allele exhibit lower serum levels of IL26. Consistently, peripheral blood mononuclear cells (PBMCs) from these patients display a reduced ability to produce IL26 and to kill *E. coli* in vitro (Piñero et al, 2017). However, the gut microbiota composition of these specific patients remains uncharacterized. Investigating whether these patients exhibit dysbiosis, particularly with increased levels of *Enterobacteriaceae*, would be of interest. Notably, we observed that epithelial homeostasis in *il26⁻/⁻* guts can be restored by the transfer of microbiota from WT to *il26⁻/⁻* larvae, highlighting the therapeutic potential of microbiota transfer in genetically predisposed IBD patients.

The prevalence of IBD has been increasing in the young population (Zhang et al, 2023), emphasizing the critical role of a balanced gut ecosystem from early life. Our findings suggest that IL-26 deficiency can influence microbiota composition and disrupt gut homeostasis during early life. These results pave the way for investigating whether early-life IL-26 levels could serve as predictive markers for IBD, offering a significant diagnostic advantage. Furthermore, future research could explore the potential of IL-26 supplementation in at-risk young individuals as a preventive strategy against IBD.

Our findings establish that IL-26 protects the gut from bacterial infections, and that this correlates with increased bacterial loads and DNA damage as well as an impaired immune response. This study underscores the potential of using IL-26 as an agent to fight off bacterial infections in the gut.

ILCs are known to regulate gut homeostasis through cytokine production in the adult zebrafish gut (16). However, the emergence of ILCs in the gut during early life and their cytokine production

profile were previously unknown. We report that functional ILCs appear in the larval gut as early as 5 dpf as the major source of IL-26. This study underscores the importance of ILCs and the cytokines they produce in gut homeostasis during the first steps of gut microbial colonization in early life.

In summary, our findings reveal key mechanisms by which host-microbiota interactions during early development, mediated by ILC-produced IL-26, protect against dysbiosis, excessive cell proliferation, and DNA damage in epithelial cells.

# Methods

**Reagents and tools table**

| Reagent/Resource | Reference or Source | Identifier or Catalog Number |
|---|---|---|
| **Experimental models** | | |
| Zebrafish | AB | NA |
| **Recombinant DNA** | | |
| **Antibodies** | | |
| anti-Mpx | GeneTex | GTX128379 |
| Anti-Rabbit IgG–Peroxidase antibody | Sigma-Aldrich | GTX128379 |
| Histone H2A.XS139ph (phospho Ser139) antibody (anti- γH2AX) | GeneTex | GTX127342 |
| 2F11 antibody | Abcam | ab71286 |
| Goat Anti-Rabbit IgG H&L | Abcam | ab175471 |
| Donkey Anti-Rabbit IgG H&L | Abcam | ab150075 |
| **Oligonucleotides and other sequence-based reagents** | | |
| qPCR primers | This study | Dataset EV7 |
| sgRNA_il26KO_1 | GCAGGGATTTATGGATGTCC | IDT |
| sgRNA_il26KO_2 | GAGACAATAAACCCTTCCAT | IDT |
| sgRNA_il20raKO_1 | TGGACGTCTCGCGGCTCAGG | IDT |
| sgRNA_il20raKO_2 | GTGAAGTGGACGGCAGGACA | IDT |
| Genotype_il26KO_Fwd | GTCAAAAGTGAGGTTGTGGCA | Eurofin |
| Genotype_il26KO_Rev | CCATGAATGCAGCCTTCAGC | Eurofin |
| Genotype_il20raKO_Fwd | GTTGTGGCTGCTGTACGCTA | Eurofin |
| Genotype_il20raKO_Rev | GGAACAGGGTTGGGAAGCTAAA | Eurofin |
| 16s_rRNA-seq_V3 | CCTACGGGNGGCWGCAG | |
| 16s_rRNA-seq_V4 | GACTACHVGGGTATCTAATCC | |
| **Chemicals, enzymes and other reagents** | | |
| Alt-R™ S.p. Cas9 Nuclease V3 | IDT | 1081058 |
| FITC-dextrane | Sigma-Aldrich | FD4 |
| Tricaine | Sigma | A5040 |
| Proteinase K | Invitrogen | 25530-049 |
| Recombinant zebrafish IL-26 protein | Kingfisher Biotech | RP1773Z |
| Single Cell RNA Purification Kit | Norgen | 51800 |
| Agilent High Sensitivity DNA Kit | Agilent | 5067-4626 |
| Click-iT™ EdU Cell Proliferation Kit | Invitrogen | C10337 |
| Click-iT™ EdU Cell Proliferation Kit | Invitrogen | C10340 |
| Click-iT™ EdU Cell Proliferation Kit | Invitrogen | C10638 |
| ProLong™ Gold Antifade Mountant | ThermoFisher | P36931 |
| DNeasy PowerSoil kit | Qiagen | 47014 |
| LongAmp Taq DNA polymerase | NEB | M0323S |
| M-MLV Reverse Transcriptase Kit | Invitrogen | 28025013 |
| Takyon™ Kit | Takyon | UF-LPMT-C0701 |
| **Software** | | |
| GraphPad Prism | https://www.graphpad.com | |
| ImageJ | https://imagej.nih.gov/ij/index.html | |
| Geneious | https://www.geneious.com | |
| **Other** | | |

## Zebrafish lines and husbandry

The zebrafish lines: wild-type (AB), *TgBAC(cldn15la-GFP)* (Alvers et al, 2014), *rag1^-/-* (Wienholds et al, 2002), *myd88^-/-* (Van Der Vaart et al, 2013), *il2rga^-/-prkdc^-/-* (Yan et al, 2019), *il26^-/-*, and *il20ra^-/-* were reared and kept in the zebrafish core facility at the Institut Curie animal facility in accordance with European Union regulations on laboratory animals using protocol numbers: APAFIS#27495-2020100614519712 v14.2.2, #2019_010 and #2022-008 (approved the French Ministry of Research). Zebrafish larvae used in collaborations with labs in other countries were maintained according to the "Acta de Aprobación 004/2021" provided by the Universidad Andrés Bello, Chile. Zebrafish embryos were collected by natural spawning of adults and were kept at 28 °C in E3 water.

## Generation of *il26*- and *il20ra*-deficient zebrafish

The coding sequence for the zebrafish interleukin-26 gene (gene name: *il26*, ENSEMBL ID: ENSDARG00000045672.6) was targeted using CRISPR/Cas9 technology with two specific sgRNAs: GCAGGGATTTATGGATGTCC and GAGACAATAAACCCTTC-CAT. Interleukin-20 receptor A (gene name: *il20ra*, ENSEMBL ID: ENSDART00000043626.6) was targeted using two specific sgRNAs: TGGACGTCTCGCGGCTCAGG and GTGAAGTGGACGGCAG-GACA. One-cell stage zebrafish embryos were injected with 1 nL of

a mixture containing guide RNA (6.65 μM) and Cas9 protein (5 μM).

## Gut barrier integrity assay

Gut permeability was assessed by oral microgavage as previously described (Cocchiaro and Rawls, 2013), with minor modifications. Briefly, zebrafish larvae were anesthetized and mounted in 2% methylcellulose. Approximately 5 nL of a 1% FITC-dextrane solution (3000–5000 MW; Sigma-Aldrich, Cat. No. FD4) was administered by microgavage. After 15 min, larvae were imaged using a spinning disk confocal microscope. Fluorescence intensity was quantified using ImageJ software by measuring the signal within the intestinal region and in the somites located dorsally. A background fluorescence signal was measured from a region outside the larva and subtracted from the somite intensity. All values were normalized to the average fluorescence intensity of the control group.

## Neutrophil quantification

Neutrophils were quantified by whole-mount immunohistochemistry. Embryos were staged and fixed overnight at 4 °C in 4% paraformaldehyde (PFA) in phosphate-buffered saline (PBS). Following fixation, embryos were washed three times for 5 min in PBS, once for 1 h in distilled water, incubated for 7 min in acetone at –20 °C, and washed again—once in distilled water and twice for 5 min in PBS containing 0.1% Tween-20 (PBST). Samples were then blocked in a solution containing 20% lamb serum, 1% dimethyl sulfoxide (DMSO), and 0.1% Tween-20 in PBS for 1 h at room temperature. Embryos were incubated overnight at 4 °C with a rabbit polyclonal anti-Mpx antibody (GeneTex, Cat. No. GTX128379) diluted 1:300 in blocking solution. After washing four times for 25 min each in PBST, samples were incubated for 30 min in blocking solution and then overnight at 4 °C with a peroxidase-conjugated anti-rabbit IgG secondary antibody (Sigma-Aldrich, Cat. No. A8275) diluted 1:100 in blocking solution. The following day, embryos were washed four times for 20 min in PBST and incubated in peroxidase substrate solution (ImmPACT DAB, Vector Laboratories, Cat. No. SK-4105) for 5 min to visualize staining. After staining, embryos were rinsed three times in PBS. Quantification of intestinal neutrophils was performed by counting stained cells in the midgut region using light microscopy.

## Genotyping

Adult zebrafish were anesthetized with tricaine (100 μg/ml, Sigma, #A5040), their tails were cut and incubated for 1 h at 56 °C with FinClip buffer (10 mM Tris, pH 8.0, 10 mM EDTA, 200 mM NaCl, 0.5% SDS) containing Proteinase K (0.2 mg/mL, Invitrogen, #25530-049). DNA was precipitated by adding 70% ethanol. The pellet was resuspended in water and the product solution was used for genotyping. Zebrafish larvae were anesthetized with tricaine, their tails were cut and incubated during 15 min at 95 °C in Base buffer (25 mM KOH, 0.2 mM EDTA). An equal volume of Neutralization buffer (40 mM Tris-HCl) was then added, and the solution was used for genotyping. *il26*[-/-] fish were genotyped by gel electrophoresis using the primers: GTCAAAAGTGAGGTTGTGGCA and CCATGAATGCAGCCTTCAGC. *il20ra*[-/-] fish were genotyped by sequencing using the primers GTTGTGGCTGCTGTACGCTA and GGAACAGGGTTGGGAAGCTAAA.

## Zebrafish Il26 protein injections

Injections were performed using 2 μL of recombinant zebrafish IL-26 protein (1 mg/mL, Kingfisher Biotech, #RP1773Z-025) mixed with 0.5 μL of phenol red. BSA at 1 mg/mL was used as a control.

## Bulk RNA-sequencing

10–15 guts per replicate were dissected and RNA was extracted using the Single Cell RNA Purification Kit (Norgen, #51800) following the manufacturer's instructions. RNA integrity and concentration were analyzed on the Agilent 4200 Tapestation system using the High Sensitivity RNA ScreenTape Analysis kit (Agilent, #5067-5579). RNA sequencing libraries were prepared from 500 ng to 1 μg of total RNA using the Illumina TruSeq Stranded mRNA Library Preparation Kit. cDNA quality was checked on the Agilent 2100 Bioanalyzer using the Agilent High Sensitivity DNA Kit (Agilent #5067-4626). After quality control, libraries were sequenced with 100-bp paired-end (PE100) reads on the NovaSeq 6000 (Illumina) sequencer. Raw data were checked for quality using FastQC (v0.11.8) and aligned to the reference genome for *Danio rerio* danRer11 from the Genome Reference Consortium. Analysis was performed in R using the EdgeR (Robinson et al, 2010) and ClusterProfile (Yu et al, 2012) packages.

## Immunostaining on dissected larval guts

Larvae were incubated with 100 μM EdU in E3 for 3 h. After incubation, larvae were washed with E3. Next, the guts were dissected and fixed in 4% paraformaldehyde for 1 h at room temperature. Samples were washed twice with PBST (0.1% Triton-X100 in PBS), followed by incubation in PBS with 3% BSA for 1 h. After one wash with PBS, the samples were incubated overnight at 4 °C with primary antibodies diluted in 200 μl of PBS: γH2AX (1:200, GeneTex, #GTX127342) or 2F11 (1:200, Abcam, #ab71286). The next day, the samples were washed three times for 10 min each in PBST. They were then incubated with secondary antibodies diluted in 500 μl of PBS at 4 °C for 3 to 4 h: Goat Anti-Rabbit IgG H&L (1:500, Abcam, #ab175471) or Donkey Anti-Rabbit IgG H&L (1:500, Abcam, #ab150075). Following this, the samples were washed twice for 10 min in PBST, and then washed three more times for 10 min in PBS. The Click-iT™ EdU Cell Proliferation Kit (Invitrogen, #C10337, #C10340, #C10638) was used according to the manufacturer's instructions. Samples were washed twice in PBS for 10 min. The dissected guts were then mounted with ProLong™ Gold Antifade Mountant (ThermoFisher, #P36931). Images were acquired using the Upright Spinning Disk Confocal Microscope (Roper/Zeiss) with a 63X objective (63x/1.4 OIL DICII PL APO, 420782-9900). Multi-dimensional imaging of the posterior gut was performed to encompass the entire intestinal tube. Quantification was carried out in ImageJ. Representative images show a single z-plane.

## Gut length analysis in larvae

Larvae were anesthetized with tricaine and mounted in 3% methylcellulose for live imaging. Gut length was measured from the intestinal bulb to the end of the intestine at the anal pore. The analysis was done in ImageJ.

## Adult body length measurements

Adult fish were anesthetized with tricaine. Body length was measured from the head to the tail, excluding the tail fin using a ruler.

## Single-cell RNA-sequencing

10 guts from 5 dpf larvae were dissected and placed in 200 μL of a dissociation cocktail (1 mg/mL fresh collagenase A, 40 μg/mL proteinase K, and 0.25% trypsin in PBS) at 37 °C for 30 min. 50 guts were dissected per condition for a total of 5 replicate tubes per condition. Cells were centrifuged at 300 RCF for 15 min at 4 °C. The pellet was resuspended in 200 μL PBS with 0.04% non-acetylated BSA. Replicates were pooled and filtered through a 40 μm cell strainer (Fisherbrand, #22363547). Samples were centrifuged at 300 RCF for 15 min at 4 °C and resuspended with in 50 ul of PBS with 0.04% non-acetylated BSA. To isolate live cells, an OptiPrep™ Density Gradient Medium (Sigma, #D1556-250ML) was used. A 40% (w/v) iodixanol working solution was prepared by mixing 2 volumes of OptiPrep™ with 1 volume of 0.04% BSA in 1X PBS/DEPC-treated water. This working solution was used to create a 22% (w/v) iodixanol solution in the same buffer. The cell suspension was mixed with one volume of the working solution and 0.45 volume of the cell suspension via gentle inversion, transferred to a 15 mL conical tube, and topped up to 6 mL with the working solution. This solution was overlaid with 3 mL of the 22% iodixanol, and then with 0.5 mL of PBS with 0.04% BSA. Samples were centrifuged at 800 RCF for 20 min at 20 °C. Viable cells were collected from the top interface, diluted in PBS with 0.04% BSA, and centrifuged at 300 RCF for 10 min at 4 °C. The supernatant was discarded, and the cells were resuspended in PBS with 0.04% non-acetylated BSA to reach the desired concentration. Cells were loaded onto a 10× Chromium instrument (10× Genomics) and libraries were prepared using the Single Cell 3′ Reagent Kit (V2 chemistry) (10× Genomics) according to the manufacturer's instructions. The sequencing coverage was approximately 100,000 reads per cell. Data analysis was performed using the Seurat package in R (Hao et al, 2024).

## Gel migration assay to determine DNA-binding ability of IL-26

Genomic zebrafish DNA was mixed with different cytokines at a final concentration of 3 ng/μl. Samples were loaded onto a 1.5% agarose gel for electrophoretic migration.

## Antimicrobial assay

The following bacterial strains were used: *Pseudomonas aeruginosa*, *Escherichia coli*, *Enterococcus faecalis*, and *Edwarsiella tarda*. Bacteria were cultured at 37 °C overnight in trypticase soy broth with 10 mM NaCl. Samples were subcultured for an additional 3 h to achieve mid–logarithmic phase growth. Bacterial concentrations were measured by spectrophotometry at 620 nm and diluted to a final concentration of $10^5$ CFU/ml, after which they were incubated at 37 °C for 24 h under low-ionic-strength conditions (10 mM NaCl). Human or zebrafish IL-26 were added to these cultures to test their antimicrobial activity. After 24 h, serial dilutions of bacterial cultures were plated onto lysogeny broth (LB) agar plates overnight. The number of colonies was counted manually.

## Generation of germ-free larvae and co-housing experiments

Fertilized zebrafish eggs were treated with bleach (0.05%) for a maximum of 2 min at 3 to 4 h post fertilization and then washed twice with sterile E3 medium for 5 min. Embryos were incubated in chlorine hypochlorite (0.003%) for 20 min. After washing, embryos were transferred to sterile E3 medium containing ampicillin (200 μg/mL), kanamycin (5 μg/mL), ceftazidime (200 μg/mL), and chloramphenicol (20 μg/mL) and placed at 28 °C in isolated containers. The medium was renewed daily under sterile conditions until the day of sample collection. Sterility was monitored every 2 days by incubating fish water in TBS media for 24 h at 37 °C. Co-housing experiments were performed by transferring GF larvae to containers with CV larvae, separated by a sterile strainer (Fisherbrand, #22363547).

## Microbial DNA extraction and 16S rRNA sequencing

20 guts from WT and $il26^{-/-}$ 5 dpf larvae per replicate were dissected. Bacterial DNA was isolated using the DNeasy PowerSoil kit (Qiagen, #47014). Amplification and barcoding was performed using V3-V4 primers (V3: CCTACGGGNGGCWGCAG; V4: GACTACHVGGGTATCTAATCC) using the LongAmp Taq DNA polymerase (NEB, M0323S) using the following PCR program: 94 °C 30 s, 30x (94 °C 30 s, 54 °C 1 min, 65 °C 40 s), 65 °C 10 min, 12 °C. After PCR clean-up (Invisorb Fragment CleanUp; Invitek 1020300300) samples were sequenced using Oxford Nanopore (MinION) using the ligation sequencing kit V14 (SQK-LSK114) and R10.4.1 flow cell. The raw sequencing data (pod5 files) were basecalled to fastq files using the super accuracy model of Dorado v0.5.3. (Oxford Nanopore Technologies, 2024). Reads were demultiplexed and subsequently filtered by size, retaining those with lengths ranging from 300 to 700 base pairs. Demultiplexed reads were analyzed with Qiime2 (Bolyen et al, 2019), using DADA2 for ASV calling and the Silva database version 138 for taxonomic classification. Redundancy analysis was done with Canoco 5.15 (Braak and Smilauer, 2012) with bacterial microbiota composition (relative abundance at the family level) as response variables, after transforming with the formula log (1000*relative_abundance + 1). RDA *p*-values were determined through permutation testing (500 permutations).

## Quantitative real-time PCR

10 guts per replicate were dissected and RNA was extracted using the Single Cell RNA Purification Kit (Norgen, #51800) following the manufacturer's instructions. Synthesis of cDNA was performed using the M-MLV Reverse Transcriptase Kit (Invitrogen, #28025013). qPCR was carried out using the Takyon™ Kit (Takyon, #UF-LPMT-C0701) on a Thermo ABI ViiA 7 Real-Time PCR System (Thermo Applied Biosystems). The primers used are listed in Dataset EV7.

## *E. tarda* infections

*Edwarsiella tarda* FL60 (provided by Dr. Phillip Klesius (USDA, Agricultural Research Service, Aquatic Animal Health Research

Unit) were grown in TSB medium + tetracycline (15 µg/mL) at 28 °C overnight. A 1:100 dilution was performed, and the bacteria were grown to reach OD600 = 0.250. The bacteria were then centrifuged at 3500 RCF for 5–10 min and resuspended in E3 water to achieve OD600 = 0.250. 6 ml of the bacterial suspension in E3 were added to six larvae for 5 h at 28 °C. Larvae were washed three times with E3 water. Survival was monitored every 12 h for 3 days post-infection. Images were acquired with the THUNDER Imager Model Organism microscope (Leica). mCherry area and mean intensity were determined using ImageJ.

### Colony-forming units (CFU) analysis from larval guts

Five guts were dissected per replicate in 100 µL of PBS. The tissue was dissociated by vortexing. 1:10 dilution was prepared, and 20 µL of the diluted sample was spread onto LB-agar plates supplemented with tetracycline (15 µg/mL). Plates were incubated overnight at 37 °C, and colonies were counted manually.

### HCR™ RNA-FISH

HCR probes for zebrafish *il26* and *nitr9* were purchased from Molecular Instruments. The staining was performed according to the manufacturer's guidelines with the following optimizations: guts were dissected and fixed in 4% paraformaldehyde for 1 h at room temperature. After completing the staining protocol, the dissected guts were mounted with ProLong™ Gold Antifade Mountant (ThermoFisher, #P36931). Images were acquired using the Upright Spinning Disk Confocal Microscope (Roper/Zeiss) with a 63X objective (63x/1.4 OIL DICII PL APO, 420782-9900).

### Statistical analysis

Statistical analyses were performed using RStudio or GraphPad Prism. Specific statistical tests and significance levels are detailed in the respective figure legends.

## Data availability

The datasets and computer code produced in this study are available in the following databases: RNA-seq profiling of dissected guts from 5-day post-fertilization WT and *il26*-/- zebrafish larvae: Gene Expression Omnibus GSE300679. RNA-seq profiling of dissected guts from 5-day post-fertilization WT and *il26*-/- zebrafish larvae reared germ-free: Gene Expression Omnibus GSE300682. RNA-seq profiling of dissected guts from 5-day post-fertilization WT zebrafish larvae 1 h post injection with BSA or recombinant zebrafish IL-26: Gene Expression Omnibus GSE300683. scRNA-seq profiling of dissected guts from 5-day post-fertilization WT and *il26*-/- zebrafish larvae.: Gene Expression Omnibus GSE300684. 16s rRNA-seq data of dissected guts from 5-day post-fertilization WT and *il26*-/- zebrafish larvae. The European Nucleotide Archive PRJEB90068.

The source data of this paper are collected in the following database record: biostudies:S-SCDT-10_1038-S44318-025-00588-w.

## Peer review information

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

## Acknowledgements

The authors would like to acknowledge the members of the animal facility of the Institut Curie for zebrafish care. The authors greatly acknowledge the Cell and Tissue Imaging (PICT-IBiSA), Institut Curie, member of the national infrastructure France-BioImaging (https://ror.org/01y7vt929) supported by the French National Research Agency (ANR-24-INBS-0005 FBI BIOGEN). This work has benefited from the facilities and expertise of CELPHEDIA-TEFOR, UAR TEFOR Paris-Saclay (TEFOR Infrastructure - Investissement d'avenir-ANR-II-INBS-0014). Life Science Editors provided editing services for this manuscript. This work was supported by the Institut Curie, INSERM, and CNRS, and the grants listed below. Laboratoire d'Excellence (Labex) DEEP (ANR-11-LBX-0044, ANR-10-IDEX- 0001-02 PSL); Ville de Paris Emergence Program (2020 DAE 78); FRM amorçage (AJE201905008718); ATIP-Avenir Starting Grant R21045DS; ERC-StG Cytok-Gut 101041422; PhD fellowships from the Ministère de l'Enseignement Supérieur et de la Recherche and from the Fondation pour la Recherche Médicale (FDT202304016654). DP and JMG were funded by the French Government's Investissement d'Avenir program, Laboratoire d'Excellence Integrative Biology of Emerging Infectious Diseases (grant no. ANR-10-LABX-62-IBEID).

## Author contributions

**Yazan Salloum**: Conceptualization; Resources; Data curation; Formal analysis; Supervision; Funding acquisition; Validation; Investigation; Visualization; Methodology; Writing—original draft; Project administration; Writing—review and editing. **Gwendoline Gros**: Investigation. **Keinis Quintero-Castillo**: Investigation; Methodology. **Camila Garcia-Baudino**: Investigation. **Soraya Rabahi**: Investigation. **Akshai Janardhana Kurup**: Investigation. **Patricia Diabangouaya**: Investigation. **David Pérez-Pascual**: Investigation; Methodology. **Rodrigo A Morales Castro**: Investigation. **Jos Boekhorst**: Investigation; Methodology. **Eduardo J Villablanca**: Funding acquisition. **Jean-Marc Ghigo**: Funding acquisition. **Carmen G Feijoo**: Funding acquisition. **Sylvia Brugman**: Funding acquisition; Investigation; Methodology. **Pedro P Hernandez**: Conceptualization; Supervision; Funding acquisition; Project administration; Writing—review and editing.

Source data underlying figure panels in this paper may have individual authorship assigned. Where available, figure panel/source data authorship is listed in the following database record: biostudies:S-SCDT-10_1038-S44318-025-00588-w.

## Disclosure and competing interests statement

The authors declare no competing interests.

# Expanded View Figures

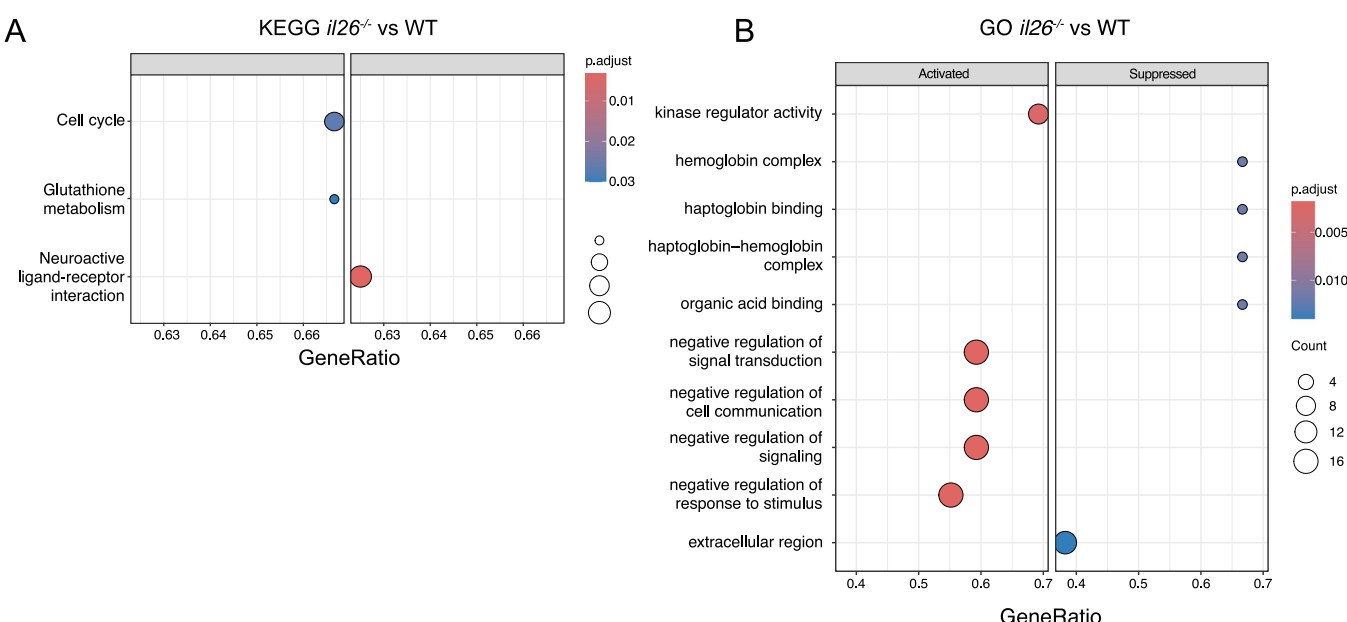

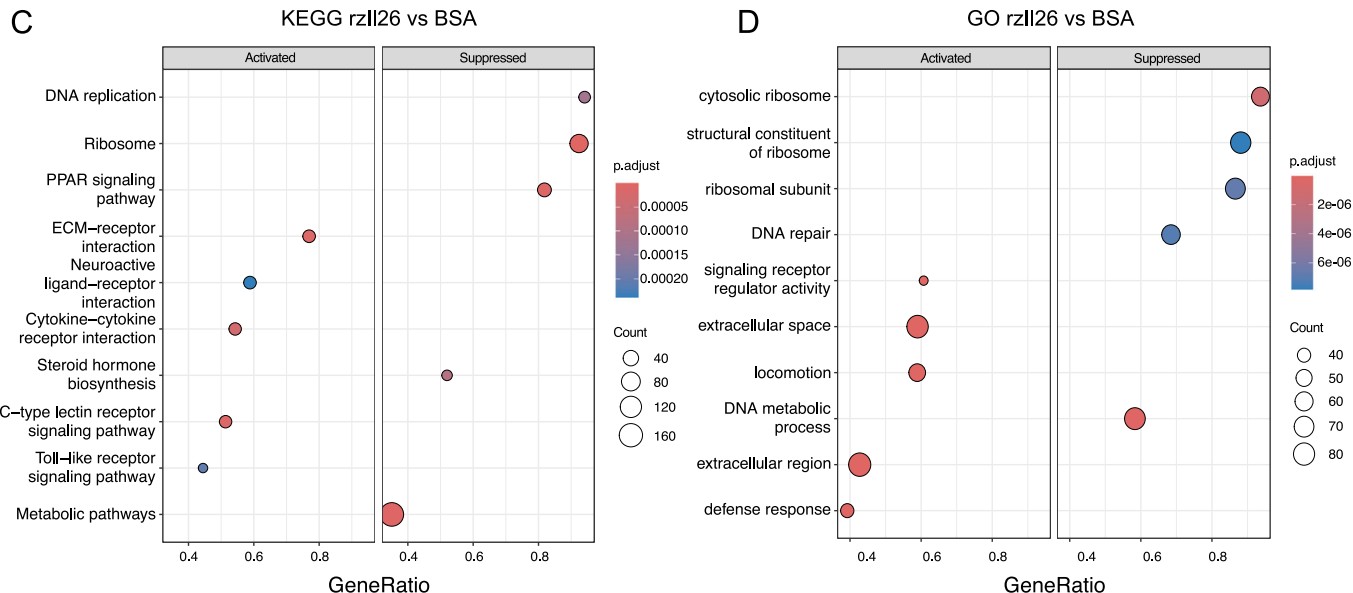

**Figure EV1.  Gene set enrichment analyses in IL-26 loss-of-function and overexpression datasets.**

(A, B) KEGG pathway (A) and GO analysis (B) in the loss-of-function dataset. (C, D) KEGG pathway (C) and GO analysis (D) in the overexpression dataset. Statistical significance was determined by Kolmogorov-Smirnov-like permutation test using the ClusterProfile package in R.

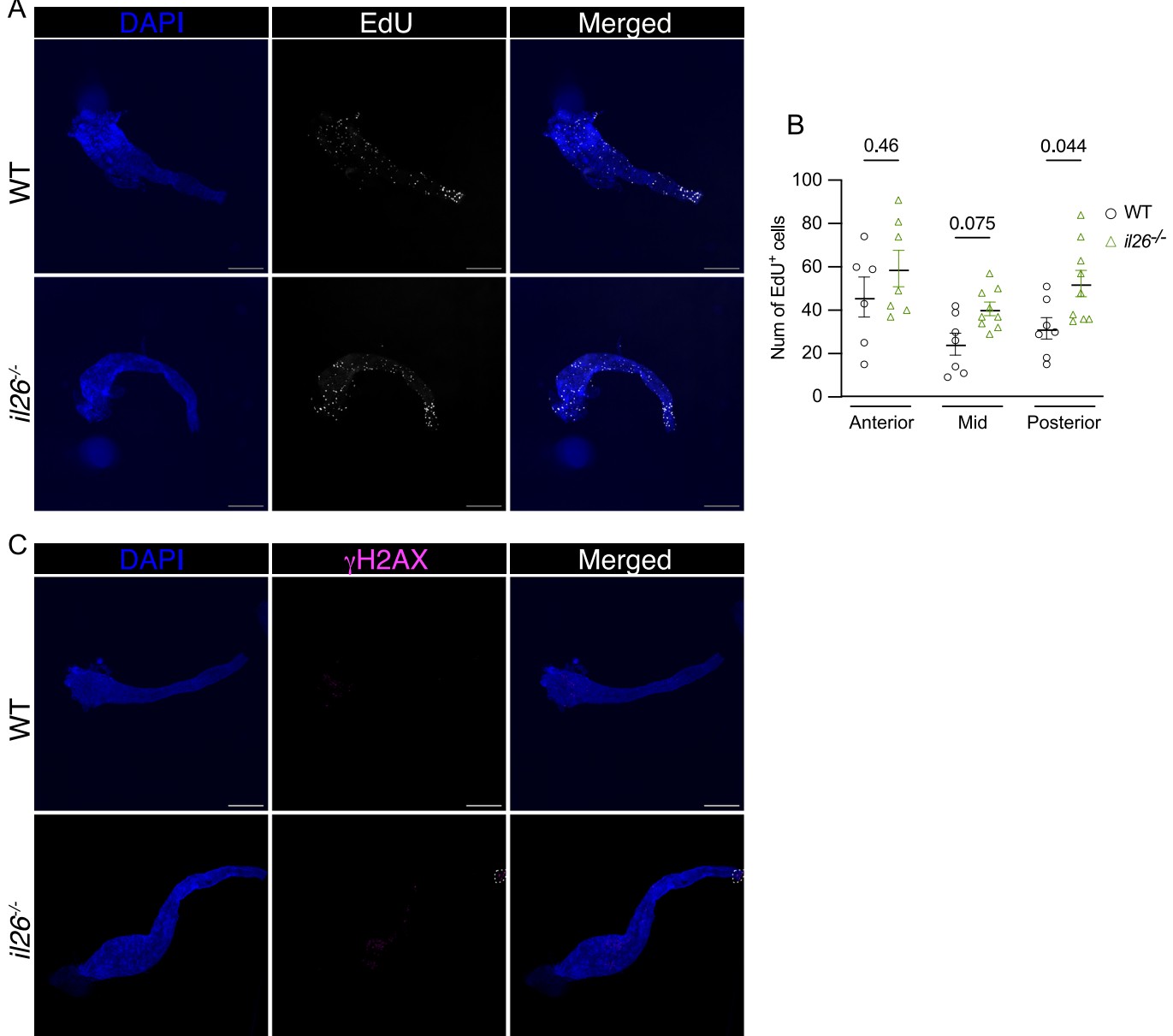

**Figure EV2. IL-26 loss increases proliferation in the mid and posterior gut and DNA damage in the posterior gut in zebrafish larvae.**

(A) Representative images of EdU staining in WT and *il26⁻/⁻* 5-dpf larval guts. Scale bars: 200 μm. (B) Quantification of EdU staining in gut segments of WT and *il26⁻/⁻* 5-dpf larvae. (C) Representative images of γH2AX staining in WT and *il26⁻/⁻* 5-dpf larval guts. Scale bars: 200 μm. Data information: (B) Data are presented as mean ± SEM. Sample sizes were as follows ((B): $n_{WT} = 7$, $n_{il26-/-} = 8$). n represents the number of biological replicates (minimum of 3 independent experiments). Statistical significance was determined by Mann–Whitney test (B). Source data are available online for this figure.

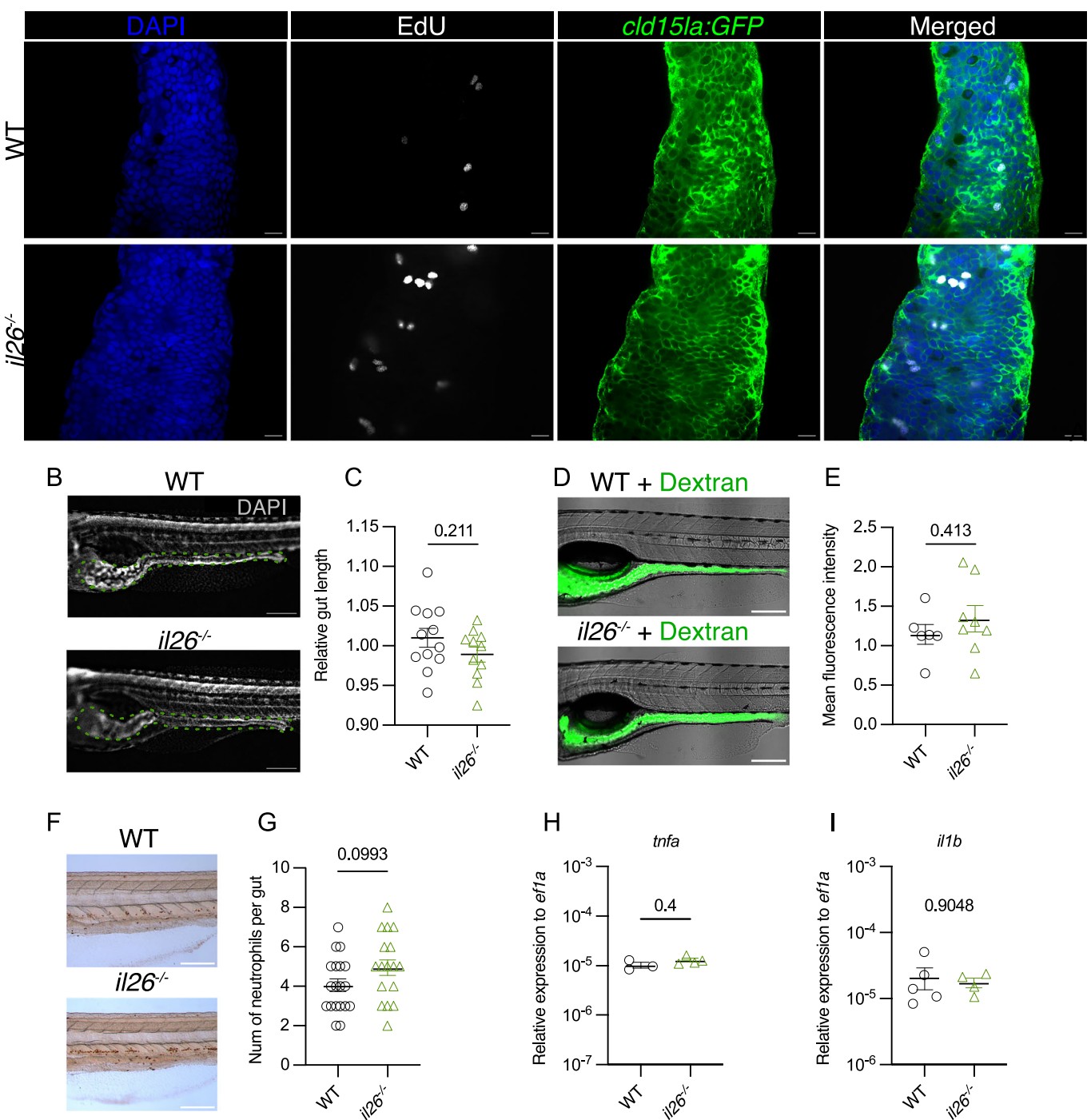

**Figure EV3.  Characterization of *il26⁻/⁻* zebrafish larvae.**

(A) EdU staining of 5-dpf *TgBAC(cldn15la-GFP)* larvae in WT and *il26⁻/⁻* genetic backgrounds. Scale bars: 10 µm. (B, C) Images of WT and *il26⁻/⁻* larvae (B) and their gut length (C). Scale bars: 200 µm. (D, E) Representative images (D) of intestinal barrier integrity analysis in WT and *il26⁻/⁻* assessed by fluorescence intensity of orally gavaged FITC–dextran in the dorsal somites (E). Scale bars: 200 µm. (F, G) Representative images (F) and quantification (G) of neutrophil staining in WT and *il26⁻/⁻*. Scale bars: 100 µm. (H, I) qRT-PCR analysis of *tnfa* (H) and *il1b* (I) in dissected guts of WT or *il26⁻/⁻*. Data information: (C, E, G, H) Data are presented as mean ± SEM. Sample sizes were as follows ((C): $n_{WT}$ = 12, $n_{il26-/-}$ = 11; (E): $n_{WT}$ = 6, $n_{il26-/-}$ = 8; (G): $n_{WT}$ = 19, $n_{il26-/-}$ = 18; (H): $n_{WT}$ = 3, $n_{il26-/-}$ = 4; (I): $n_{WT}$ = 5, $n_{il26-/-}$ = 4). n represents the number of biological replicates (minimum of 3 independent experiments). Statistical significance was determined by Mann–Whitney test (C, E, G, H). Source data are available online for this figure.

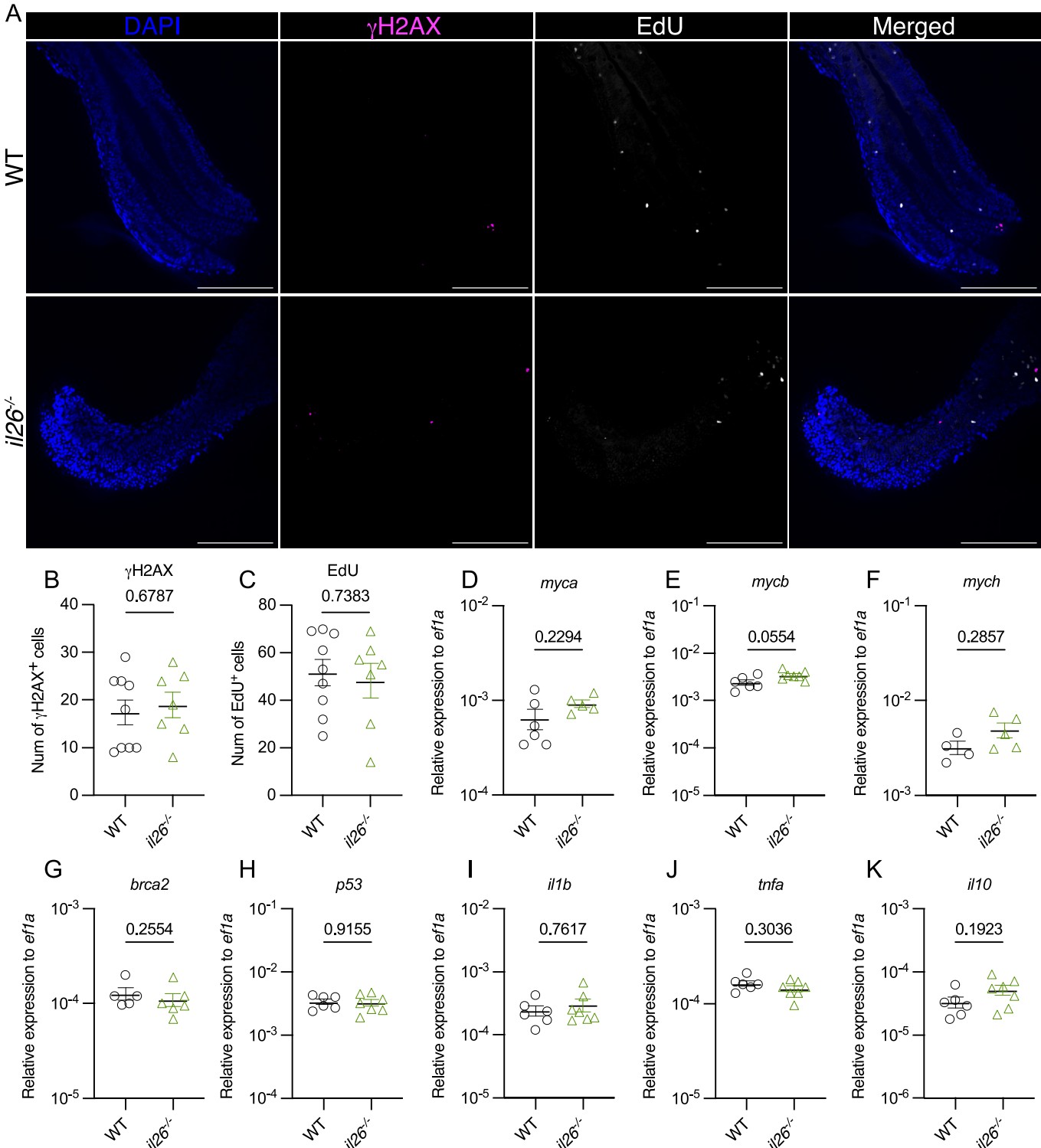

**Figure EV4. Characterization of *il26-/-* juvenile zebrafish.**

(A) Representative images of γH2AX and EdU staining in WT and *il26-/-* 5-wpf guts. Scale bars: 100 μm. (B, C) Quantification of γH2AX (B) and EdU staining (C) in WT and *il26-/-* 5-wpf posterior guts. (D–K) qRT-PCR analysis of selected genes in dissected guts of WT or *il26-/-* 5-wpf fish. Data information: (B–K) Data are presented as mean ± SEM. Sample sizes were as follows ((B, C): $n_{WT}$ = 9, $n_{il26-/-}$ = 7; (D–K): $n_{WT}$ = 4-6, $n_{il26-/-}$ = 5-7). n represents the number of biological replicates (minimum of 3 independent experiments). Statistical significance was determined by Mann–Whitney test (B–K). Source data are available online for this figure.

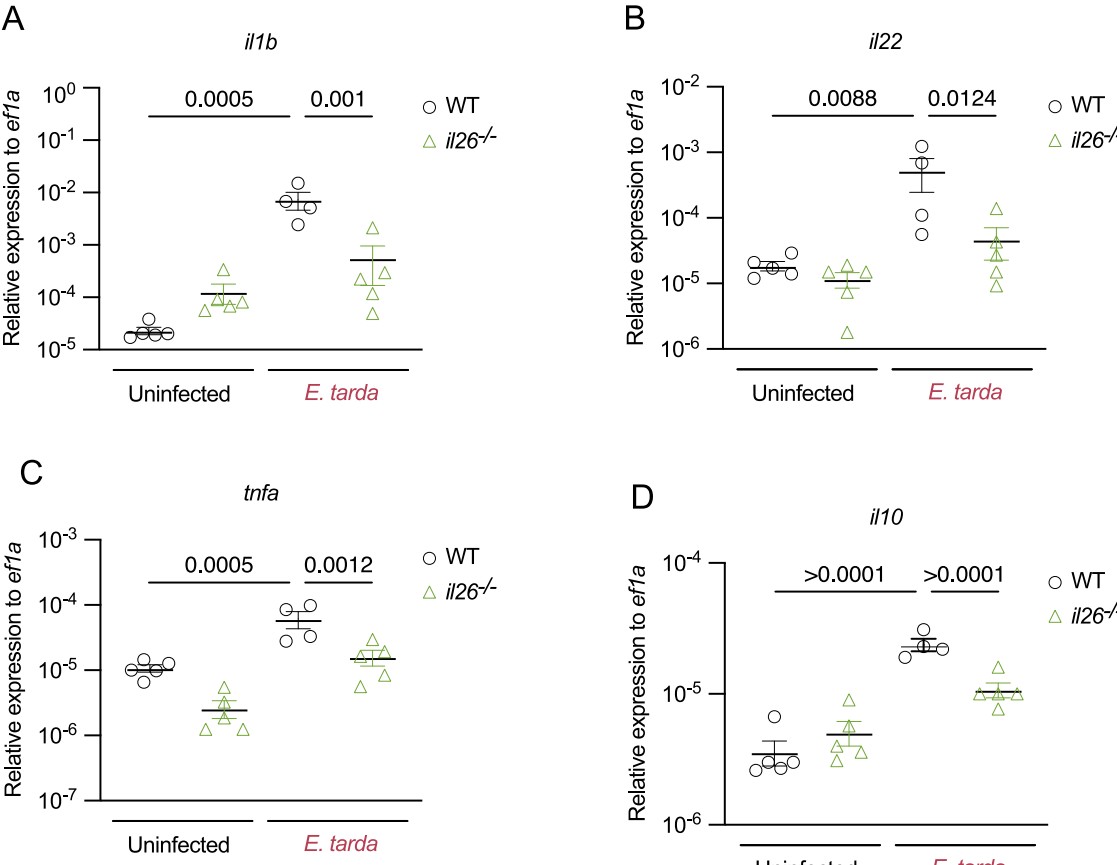

**Figure EV5.  Cytokine expression in WT and *il26⁻/⁻* larval guts following *E. tarda* infection.**

(A, D) qRT-PCR analysis of *il1b* (A), *il22* (B), *tnfa* (C), *il10* (D) in dissected guts of WT and *il26⁻/⁻* at 3 dpi. Data information: (A–D) Data are presented as mean ± SEM. Sample sizes were as follows ((A–D): $n_{WT\ uninfected} = 5$, $n_{WT\ infected} = 4$, $n_{il26\text{-/-}\ uninfected} = 5$, $n_{il26\text{-/-}\ infected} = 5$). n represents the number of biological replicates (minimum of 3 independent experiments). Statistical significance was determined by one-way ANOVA (A–D). Source data are available online for this figure.

