## [Peer Review File · The EMBO Journal]

IL-26 from innate lymphoid cells regulates early-life gut epithelial homeostasis by shaping microbiota composition

Yazan Salloum, Gwendoline Gros, Keinis Quintero-Castillo, Camila Garcia-Baudino, Soraya Rabahi, Akshai Janardhana Kurup¹, Patricia Diabangouaya, David Pérez-Pascual, Rodrigo Morales, Jos Boekhorst, Eduardo Villablanca, Jean-Marc Ghigo, Carmen Feijoo, Sylvia Brugman, and Pedro Pablo Hernandez

Corresponding author: Pedro Pablo Hernandez (phernand@curie.fr)

Review Timeline:

Submission Date:	11th Mar 25
Editorial Decision:	11th Apr 25
Revision Received:	8th Jul 25
Editorial Decision:	18th Aug 25
Revision Received:	2nd Sep 25
Accepted:	16th Sep 25

Editor: Ieva Gailite

Transaction Report:

Dear Pedro,

Thank you for submitting your manuscript for consideration by the EMBO Journal. We have now received comments from a full set of reviewers, which are included below for your information.

As you will see, all reviewers are generally positive in their assessment and appreciate the contribution of the study to the research field. At the same time, they indicate several concerns that would be important to address in the revised study. From my side, I find these points generally reasonable. Therefore, I invite you to address these comments in a revised manuscript. I think that it would be useful to discuss the revision in more detail via email or phone/videoconferencing - please let me know which option you prefer.

We generally allow three months as standard revision time, which can be extended to six months in the case of major revisions. Should you foresee a problem in meeting this deadline, please let us know in advance to discuss an extension. As a matter of policy, competing manuscripts published during this period will not negatively impact on our assessment of the conceptual advance presented by your study. However, please contact me as soon as possible upon publication of any related work to discuss the appropriate course of action.

When preparing your letter of response to the referees' comments, please bear in mind that this will form part of the Review Process File and will therefore be available online to the community. For more details on our Transparent Editorial Process, please visit our website: <https://www.embopress.org/page/journal/14602075/authorguide#transparentprocess>. Please also see the attached instructions for further guidelines on preparation of the revised manuscript.

Please feel free to contact me if you have any further questions regarding the revision. Thank you for the opportunity to consider your work for publication. I look forward to your revision.

With best wishes,

leva

leva Gailite, PhD
Senior Scientific Editor
The EMBO Journal
Meyerohofstrasse 1
D-69117 Heidelberg
Tel: +4962218891309
i.gailite@embojournal.org

We realize that it is difficult to revise to a specific deadline. In the interest of protecting the conceptual advance provided by the work, we recommend a revision within 3 months (10th Jul 2025). Please discuss the revision progress ahead of this time with the editor if you require more time to complete the revisions.

Referee #1:

Overall summary

How the host immune system and microbiota interact to maintain gut homeostasis is the subject of intense investigation; identifying factors that regulate early life microbiota composition can significantly impact human health.

In this study authors combine state of the art zebrafish genetics, transcriptomics (RNAseq, single cell), and microscopy with microbiota profiling (not common in zebrafish infection field), gnotobiotics (engineering microbiota composition) and innovative bacterial infection models to reveal fundamental roles for IL26 in gut homeostasis. Results show that IL26 regulates microbiota composition (cell proliferation, DNA damage), independent of IL26 receptor (bacterial killing), and that ILCs are a primary source of IL26. Together, findings highlight IL26 as key cytokine linking microbiota, ILCs, and intestinal epithelial cells to maintain gut homeostasis.

This high quality work is an inspirational example of exploiting zebrafish models to address questions paramount to human health. It is exciting to consider these mechanisms and how they protect against dysbiosis. I am highly enthusiastic for this manuscript to advance to publication, I only have comments out of general interest.

Comments authors may consider in future work

1. How is IL26 antibacterial, is it pore formation? How is it specific to only some bacterial species? Is it possible that zebrafish IL26 acts better against fish-adapted pathogens? In contrast to human IL26 the zebrafish protein did not bind to DNA, why not? Any clues from sequence alignment?
2. Future in depth characterisation / imaging of the IL26 KO zebrafish would be of great interest. Does IL26 KO impact the gut in other non-microbiota ways? eg gut motility, loss of mucosal barriers, mucin production by goblet cells, morphological changes in gut epithelial structure.
3. To the authors' credit this results have therapeutic implications that warrant consideration
 - Development of IL26 supplementation for IBD patients with elevated levels of Enterobacteriaceae
 - Microbiota transfer in genetically predisposed IBD patients
 - Use of IL26 to control gut bacterial infectionsIt will be a great pleasure to follow the future of this research avenue.

Minor

1. Line 260. Reference for this statement?

Non essential suggestions

The authors present a first in vivo animal model to study impact of IL26 on gut homeostasis.

A related paper by Hernandez (Science Immunology 2018) discovering ILCs in adult zebrafish was transformative for the field. Here authors first discovered ILCs (showing cell type diversity resembling that of human ILCs) are present in the zebrafish gut. This new report goes significantly beyond this initial discovery, demonstrating in vivo consequences of IL26 / ILCs on gut microbiota and epithelial homeostasis in early life.

Overall, this is a tour de force effort. I hope this manuscript attracts the attention it deserves among EMBO readership.

Referee #2:

Salloum et al generate il26 and il20ra mutant zebrafish to study the role of IL-26 in intestinal physiology. They find a microbiota-dependent role for IL-26 in suppressing DNA damage in IECs and intestinal stem cell proliferation. The DNA damage marker was also found to be responsive to IL-26 deletion in an *E. tarda*-induced DNA damage challenge. The manuscript concludes with identification of il7r/orc/nitr4a positive ILCs as the source of IL-26 in the zebrafish larval gut.

The study provides important new functional analysis of IL-26 in intestinal homeostasis (I have a minor note below about the author's "first" claim). I like the significance of the study, the questions and techniques are well devised.

Major concerns

1. Please clarify the scheme to generate experimental zebrafish. From line 240, it sounds like homozygous mutant adults were used to generate -/- embryos rather than a more conventional heterozygote adult in cross. Where clutchmate WT adults raised under the same conditions/held on the same system, but maintained in separate tanks used to generate the comparison WT embryos?

2. Figure 3 co-housing experiment: please clarify the timing of this experiment, ie when did co-housing start and what was the input for conventionalisation in the methods (line 506) or results (line 240). It is currently unclear how long the WT microbiota is interacting with the il26 mutant and the length of time that the host has to select microbiota. Why is the WT group missing from the co-housed analyses? There are some images in the supplementary figure.

Does the reciprocal co-housing experiment where GF WT embryos are introduced into il26 mutant-conditioned media prove the presence of a pathobiont microbiota? It would be equally interesting if the WT embryos are able to deplete the pathobiont components of the il26 mutant microbiota (presumably through IL26 IBA) and do not develop pathology.

3. Supplementary Figure 4: The il120ra knockout is nice but there is no data to demonstrate that it is unresponsive to rzIL26 administration (could read out top RNAseq hits by qPCR, ideally match to something functional like EdU stain #s) to confirm that this is the cognate receptor.

Minor concerns

Supplementary Figure 1: can the authors provide immune phenotyping data to show that there is no colitis phenotype?

Figure 1 E-G should be repeated with the rzIL26 administration. This is important to determine if the loss of DNA damage transcription response is due to anergy to baseline damage or if IL26 actively suppresses γ H2AX-labeled DNA damage. These data would allow expansion of the discussion in synergy with the later microbiome work.

Line 171: please specify what the 2F11 antigen is supposed to mark.

Supplementary Figure 5A: the intensity of the zIL26 10 uM band is clearly less than the bands either side. How do the authors reach their conclusion without more precise quantification?

Figure 4C-D: please insert in vitro rzIL26 vs *E. tarda* CFU data here. This is necessary to connect endogenous IL26 Fig 3 microbiota effects to the infection experiment.

Line 320 is very very specific for "homeostasis". Zeng et al recently published a paper showing IL-26 protects the carp gut against *A. hydrophilia* damage. I suggest removing the word "first".

Referee #3:

This manuscript by Salloum Y. et al. used the zebrafish model to investigate the role of IL-26 in regulating intestine epithelium cell physiology in the larvae stage. It showed that IL-26, by shaping the gut microbial community, promotes intestine epithelium cell proliferation and DNA damage. Pathogenic bacteria infection upregulates IL-26 expression, which enhances host survival. Overall, this manuscript incorporated diverse methods and presented exciting data that will advance our understanding of the interactions among the immune system-intestine epithelium cells-microbiota. Please find below several major and minor concerns associated with this manuscript.

Major concerns:

(1) The expression of *il26* at the basal level is extremely low. As the author pointed out, it is not expressed in their single-cell RNA seq data sets. By searching two other zebrafish single-cell RNA seq datasets (Farnsworth et al. 2019. *Dev Biol. & Sur A. et al.*, 2023 *Development Cell*), the expression of *il26* also can't be detected. The extremely low expression of *il26* is also shown by the real-time PCR results in Figures 4 and 5.

It is unclear how such an extremely lowly expressed protein induces profound effects in the gut microbial community and intestine epithelium cell physiology at the basal level.

Can the authors provide more convincing pieces of evidence to show that *il26* is expressed in zebrafish larvae at the basal level? For example, can the author show the *il26* ISH image of the whole zebrafish larvae with negative staining control so that it will be better appreciated than the signal revealed in Figure 5E? Can the authors also present data to show at which developmental stage *il26* is expressed in zebrafish larvae?

(2) Can the author also provide evidence to show whether IL-26 is also expressed in other immune cells besides ILCs? Within ILCs, is IL26 expressed in a specific ILC subgroup?

(3) Can the author also provide some phenotypes concerning the adult *il26*^{-/-} zebrafish to show the consequence of increased intestine epithelium cell proliferation and DNA damage observed in the embryonic stage? It is expected that the change in the gut microbial community and intestine epithelium cell phenotype during the embryonic stage due to the lack of *il26* may be exacerbated when the zebrafish continue to grow and develop. For example, is there an increased inflammation in *il26*^{-/-} adult zebrafish?

(4) The authors indicate that the effects of *il26* on intestine epithelium proliferation and DNA damage are cell-type and intestine region-specific. It seems in most figures that the increased DNA damage was quantified in the very distal region of the intestine. However, in Figure 2, shows that DNA damage signal is mostly enriched in *fabp6* high cells. In zebrafish, *fabp6* is enriched in a small cell population that resembles the ileum (Wen J. et al., *Science Advances*, 2021), which is in the middle of the intestine. This seems to be contrary to the IF staining data. Can the author show staining evidence to further prove the intestine region-specific and the cell type-specific response induced by *il26*^{-/-}? In addition, can the author explain why such an intestine region/cell type-specific response is exhibited? For example, is there an enriched innate immune cell in the zebrafish ileum region in *il26*^{-/-}?

Minor concerns:

(1) Figure 1 E, both DNA and Edu are in blue color, making it extremely hard to see the Edu labeling. Consider changing the color of DNA to gray so that the other two labels can be visualized better.

(2) Figure 1 E, please also add an image of the whole zebrafish intestine to show the intestine region-specific effects.

(3) Some of the IF staining images, such as Figure S2A, are not clear. The γ H2AX is not visible in the image panels.

Referee #1:**Overall summary**

How the host immune system and microbiota interact to maintain gut homeostasis is the subject of intense investigation; identifying factors that regulate early life microbiota composition can significantly impact human health.

In this study authors combine state of the art zebrafish genetics, transcriptomics (RNAseq, single cell), and microscopy with microbiota profiling (not common in zebrafish infection field), gnotobiotics (engineering microbiota composition) and innovative bacterial infection models to reveal fundamental roles for IL26 in gut homeostasis. Results show that IL26 regulates microbiota composition (cell proliferation, DNA damage), independent of IL26 receptor (bacterial killing), and that ILCs are a primary source of IL26. Together, findings highlight IL26 as key cytokine linking microbiota, ILCs, and intestinal epithelial cells to maintain gut homeostasis. This high quality work is an inspirational example of exploiting zebrafish models to address questions paramount to human health. It is exciting to consider these mechanisms and how they protect against dysbiosis. I am highly enthusiastic for this manuscript to advance to publication, I only have comments out of general interest.

We thank the reviewer for the positive and encouraging assessment of our work. We are pleased that the significance and approach of the study were appreciated, and we value the thoughtful suggestions for future research.

Comments authors may consider in future work

1. How is IL26 antibacterial, is it pore formation? How is it specific to only some bacterial species? Is it possible that zebrafish IL26 acts better against fish-adapted pathogens? In contrast to human IL26 the zebrafish protein did not bind to DNA, why not? Any clues from sequence alignment?

We broke this comment down into several questions here below.

How is IL26 antibacterial, is it pore formation? How is it specific to only some bacterial species?

Human IL-26 contains a high number of positively charged amino acids, including 30 lysine and arginine residues. As a result, its isoelectric point is 10.7, highlighting its strongly cationic and alkaline nature. Structural predictions further revealed that IL-26 exhibits a distinct amphipathic character—with a surface enriched in cationic charges on one side and a hydrophobic patch on the other. The cationic amphipathic nature is similar to that of classical antimicrobial peptides (AMPs), suggesting that IL-26 functions similarly. These properties allow IL-26 to bind to and penetrate bacterial membranes, which are typically negatively charged, thereby killing these bacteria by pore formation. As for species specificity, it remains unclear which bacterial species are most susceptible to IL-26. It is likely that the net surface charge of bacterial membranes plays a role in susceptibility, but this has not yet been systematically studied. Based on available evidence, we hypothesize that IL-26 could exert bactericidal effects against a broad range of bacterial species, provided that sufficient

concentrations of IL-26 are reached. We have expanded our discussion to include several of these points (line 418).

Is it possible that zebrafish IL26 acts better against fish-adapted pathogens?

To our knowledge, there is no evidence of any host-pathogen adaptation of IL-26. However, this is a very interesting premise and certainly worth investigating.

In contrast to human IL26 the zebrafish protein did not bind to DNA, why not? Any clues from sequence alignment?

We would like to highlight that we have slightly revised our conclusion to better describe our finding: zebrafish IL-26 can bind DNA, but with lower affinity than the human protein (line 257; Figs. 5A,B).

This difference between human and zebrafish IL-26's ability to bind DNA was unexpected. This is because zebrafish IL-26 contains 34 positively charged residues and given that the cationic nature of human IL-26 was suggested to be implicated in its DNA binding properties, we initially assumed similar feature for zebrafish IL-26. We have discussed this with structural biologists who were interested in further addressing this question through crystallographic analysis of IL-26–DNA complexes in both species. We discuss this in line 432.

2. Future in depth characterisation / imaging of the IL26 KO zebrafish would be of great interest. Does IL26 KO impact the gut in other non-microbiota ways? eg gut motility, loss of mucosal barriers, mucin production by goblet cells, morphological changes in gut epithelial structure.

We agree that these are promising directions. We have included data showing no differences in gut barrier integrity, gut neutrophil numbers, or inflammatory cytokine expression in *il26*^{-/-} larvae (line 173; Figs. EV3D-I).

3. To the authors' credit this results have therapeutic implications that warrant consideration

- Development of IL26 supplementation for IBD patients with elevated levels of Enterobacteriaceae
- Microbiota transfer in genetically predisposed IBD patients
- Use of IL26 to control gut bacterial infections

It will be a great pleasure to follow the future of this research avenue.

We thank the reviewer for their encouraging comment and for highlighting the potential therapeutic implications of our findings. We greatly appreciate their thoughtful insights and kind words.

Minor

1. Line 260. Reference for this statement?

We have added the following references (line 321) (Yang et al, 2017; Pleguezuelos-Manzano et al, 2020; Prorok-Hamon et al, 2014).

Non essential suggestions

- The authors present a first in vivo animal model to study impact of IL26 on gut homeostasis.
- A related paper by Hernandez (Science Immunology 2018) discovering ILCs in adult zebrafish was transformative for the field. Here authors first discovered ILCs (showing cell type diversity resembling that of human ILCs) are present in the zebrafish gut. This new report goes significantly beyond this initial discovery, demonstrating in vivo consequences of IL26 / ILCs on gut microbiota and epithelial homeostasis in early life.
- Overall, this is a tour de force effort. I hope this manuscript attracts the attention it deserves among EMBO readership.

We sincerely thank the reviewer for their generous and encouraging feedback.

Referee #2:

Salloum et al generate *il26* and *il20ra* mutant zebrafish to study the role of IL-26 in intestinal physiology. They find a microbiota-dependent role for IL-26 in suppressing DNA damage in IECs and intestinal stem cell proliferation. The DNA damage marker was also found to be responsive to IL-26 deletion in an E. tarda-induced DNA damage challenge. The manuscript concludes with identification of *il7r/rorc/nitr4a* positive ILCs as the source of IL-26 in the zebrafish larval gut.

The study provides important new functional analysis of IL-26 in intestinal homeostasis (I have a minor note below about the author's "first" claim). I like the significance of the study, the questions and techniques are well devised.

We thank the reviewer for the positive assessment and appreciation of the significance of our study. We are grateful for the constructive feedback and the opportunity to clarify key aspects of our experimental design.

Major concerns:

1. Please clarify the scheme to generate experimental zebrafish. From line 240, it sounds like homozygous mutant adults were used to generate $-/-$ embryos rather than a more conventional heterozygote adult in cross. Where clutchmate WT adults raised under the same conditions/held on the same system, but maintained in separate tanks used to generate the comparison WT embryos?

We have clarified in the revised manuscript that clutchmate WT and *il26*^{-/-} adult zebrafish were used to generate the experimental embryos (line 108). These adult fish were raised under identical conditions but maintained in separate tanks. We chose this approach because using a heterozygous incross would produce mixed-genotype offspring (WT and *il26*^{-/-}) that are co-housed, potentially confounding any effects of IL-26 deficiency on the gut microbiota due to microbial exchange.

2. Figure 3 co-housing experiment: please clarify the timing of this experiment, ie when did co-housing start and what was the input for conventionalisation in the methods (line 506) or results (line 240). It is currently unclear how long the WT microbiota is interacting with the *il26* mutant and the length of time that the host has to select microbiota. Why is the WT group missing from the co-housed analyses? There are some images in the supplementary figure.

Does the reciprocal co-housing experiment where GF WT embryos are introduced into *il26* mutant-conditioned media prove the presence of a pathobiont microbiota? It would be equally interesting if the WT embryos are able to deplete the pathobiont components of the *il26* mutant microbiota (presumably through IL26 IBA) and do not develop pathology.

In the revised manuscript, we have explained that *il26*^{-/-} larvae were raised GF until 3 dpf. At that point, they were co-housed with WT CV larvae by transferring the *il26*^{-/-} larvae into WT plates, separated by a 40 µm sterile strainer up to 5 dpf. This setup allowed microbial exchange without physical mixing of the larvae (line 298).

We acknowledge the reviewer's question regarding the missing co-housed WT group. We have included this in the revised manuscript (line 303; Figs. 6E,F). These experiments showed that microbiota transfer from *il26*^{-/-} to WT does not induce pathology in WT. We agree with the reviewer that this could be due to the presence of IL-26 in WT embryos, which may help contain or eliminate pathobionts, preventing phenotype development (discussed in line 449).

3. Supplementary Figure 4: The *il120ra* knockout is nice but there is no data to demonstrate that it is unresponsive to rzIL26 administration (could read out top RNAseq hits by qPCR, ideally match to something functional like EdU stain #s) to confirm that this is the cognate receptor.

We appreciate the reviewer's comment about the importance of providing evidence that *il20ra*^{-/-} fish are unresponsive to IL-26. However, we would like to indicate that measuring gene expression after IL-26 injection in *il20ra*^{-/-} to confirm that receptor-mediated signaling is disrupted in these fish could be confounded by the receptor-independent functions of IL-26.

To avoid this issue, we analyzed *il20ra* levels, which are more likely to reflect receptor-dependent signaling of IL-26. We have included data showing that *il20ra* expression was significantly reduced at steady state in *il20ra*^{-/-} guts compared to WT, suggesting effective gene disruption. Moreover, rzIL26 injection strongly induced *il20ra* expression in WT guts, but this induction was absent in *il20ra*^{-/-}, supporting the conclusion that IL-26 receptor-mediated signaling is disrupted in *il20ra*^{-/-} (line 243; Fig. 4C).

Minor concerns:

1. Supplementary Figure 1: can the authors provide immune phenotyping data to show that there is no colitis phenotype?

We have included data showing no differences in gut barrier integrity, gut neutrophil numbers, or inflammatory cytokine expression in *il26*^{-/-} larvae (line 173; Figs. EV3D-I). These findings indicate the absence of a colitis phenotype in *il26*^{-/-} larvae.

2. Figure 1 E-G should be repeated with the rzIL26 administration. This is important to determine if the loss of DNA damage transcription response is due to anergy to baseline damage or if IL26 actively suppresses γ H2AX-labeled DNA damage. These data would allow expansion of the discussion in synergy with the later microbiome work.

We thank the reviewer for this insightful suggestion. We have addressed this in the revised manuscript (line 145).

“To determine whether IL-26 prevents DNA damage in the gut or induces unresponsiveness to baseline DNA damage, γ H2AX staining was performed on 5-dpf WT and *il26*^{-/-} larval guts 5 hours following rzIL26 injection in the gut and swim bladder. rzIL26 administration did not affect the number of γ H2AX-positive cells in either WT or *il26*^{-/-} larvae (Appendix Fig. S3A). This result argues against a rapid transcriptional role for IL-26 in causing unresponsiveness to DNA damage. Moreover, γ H2AX staining in *il26*^{-/-} appeared pan-nuclear rather than focal (Fig. 1E), a pattern typically associated with widespread DNA damage, as opposed to discrete γ H2AX foci associated with enhanced responsiveness to DNA damage. Together, these results

support the hypothesis that IL-26 plays a protective role in preventing the accumulation of true DNA lesions in the posterior gut, rather than only modulating the DNA damage response. This protective function of IL-26 cannot be recapitulated by a single high-dose administration of the protein.”

“In addition, rzIL26 injection did not alter the number of EdU-positive cells in either WT or *il26*^{-/-} larvae (Appendix Fig. S3B), suggesting that IL-26-mediated modulation of proliferation is not driven by a rapid transcriptional response. We acknowledge that our current exogenous administration model may not faithfully recapitulate the endogenous expression of IL-26. Differences in concentration, timing, and delivery route may limit its effectiveness. Therefore, the absence of an effect of IL-26 administration on EdU and γ H2AX analyses does not conclusively rule out an effect of IL-26 supplementation on these processes. Further experiments with more physiologically relevant delivery may be necessary to fully address this question.”

We have also discussed this finding considering the microbiome work (line 455). “Recombinant zebrafish IL-26 administration did not lower proliferation and DNA damage levels in either WT or *il26*^{-/-} larvae. This finding argues against a direct and rapid transcriptional role for IL-26 in modulating these phenotypes and therefore supports the hypothesis that these phenotypes are caused by dysbiosis. This is because a single exogenous administration of rzIL26 at 5 dpf may not be sufficient to influence microbiota composition, compared to constitutive endogenous expression starting from 3 dpf, when microbial colonization begins. Further studies aiming to profile microbiota composition and deliver IL-26 using more physiologically relevant methods may be necessary to fully evaluate this possibility.”

3. Line 171: please specify what the 2F11 antigen is supposed to mark.

We have added that the 2F11 antibody recognizes annexin A4 (*anxa4*) and is commonly used to label secretory cells in zebrafish larvae (223). We have also provided a dot plot from our scRNA-seq data demonstrating that *anxa4* showed minimal expression in absorptive enterocytes and higher expression in goblet cells, enteroendocrine cells (EECs), Best4⁺ ECs, and epithelial progenitors (Fig. 3B).

4. Supplementary Figure 5A: the intensity of the zIL26 10 μ M band is clearly less than the bands either side. How do the authors reach their conclusion without more precise quantification?

We thank the reviewer for the insightful comment. We have slightly revised our conclusion to better describe our finding (line 257). “In contrast to hIL26, incubating rzIL26 with genomic DNA (gDNA) at a concentration of 1 μ M did not block DNA migration (Fig. 5A). At 10 μ M, rzIL26 only partially impeded DNA migration (Fig. 5A), while at 20 μ M, a clear inhibition was observed (Fig. 5B). These results reveal a weaker DNA-binding capacity of zebrafish IL-26 compared to the human counterpart.”

5. Figure 4C-D: please insert in vitro rzIL26 vs *E. tarda* CFU data here. This is necessary to connect endogenous IL26 Fig 3 microbiota effects to the infection experiment.

We have included data showing that rzIL26 does not kill *E. tarda* in vitro (line 335; Fig. 7F). This led us to investigate the role of IL-26 receptor signaling during *E. tarda* infections using

il20ra^{-/-}. “Similar to *il26*^{-/-} larvae, *il20ra*^{-/-} mutants displayed increased susceptibility to *E. tarda* infection (Fig. 7G). These findings indicate that IL-26 exerts its protective effects in a receptor-dependent manner.” This result also constitutes another validation of *il20ra*^{-/-} fish (in connection to major comment #3).

6. Line 320 is very very specific for "homeostasis". Zeng et al recently published a paper showing IL-26 protects the carp gut against A. hydrophilia damage. I suggest removing the word "first".

We understand the legitimate reviewer concern. However, in our writing, we claim that we created the first *in vivo* animal model to study the impact of IL-26 loss-of-function on gut homeostasis. In the mentioned report by Zang et al., the *in vivo* experiments were performed with recombinant IL-26 administration, thereby our claim holds from the perspective that we provided the first knockout animal model to study IL-26 loss-of-function.

Referee #3:

This manuscript by Salloum Y. et al. used the zebrafish model to investigate the role of IL-26 in regulating intestine epithelium cell physiology in the larvae stage. It showed that IL-26, by shaping the gut microbial community, promotes intestine epithelium cell proliferation and DNA damage. Pathogenic bacteria infection upregulates IL-26 expression, which enhances host survival. Overall, this manuscript incorporated diverse methods and presented exciting data that will advance our understanding of the interactions among the immune system-intestine epithelium cells-microbiota. Please find below several major and minor concerns associated with this manuscript.

We thank the reviewer for the positive and constructive evaluation of our manuscript. We are pleased that the study approach and relevance were appreciated, and we value the opportunity to clarify the raised points.

Major concerns:

(1) The expression of *il26* at the basal level is extremely low. As the author pointed out, it is not expressed in their single-cell RNA seq data sets. By searching two other zebrafish single-cell RNA seq datasets (Farnsworth et al. 2019. *Dev Biol. & Sur A.* et al., 2023 *Development Cell*), the expression of *il26* also can't be detected. The extremely low expression of *il26* is also shown by the real-time PCR results in Figures 4 and 5.

It is unclear how such an extremely lowly expressed protein induces profound effects in the gut microbial community and intestine epithelium cell physiology at the basal level. Can the authors provide more convincing pieces of evidence to show that *il26* is expressed in zebrafish larvae at the basal level? For example, can the author show the *il26* ISH image of the whole zebrafish larvae with negative staining control so that it will be better appreciated than the signal revealed in Figure 5E? Can the authors also present data to show at which developmental stage *il26* is expressed in zebrafish larvae?

We thank the reviewer for raising this important point. We agree that the observation of a steady-state phenotype despite low basal expression levels of *il26* is initially surprising. However, this highlights an important concept: that low mRNA levels do not necessarily equate to low biological impact. This is especially relevant for cytokines, which are often expressed at very low levels under homeostatic conditions and can be tightly regulated both temporally and spatially. Such low expression makes them notoriously difficult to detect in single-cell RNA-seq datasets, as previously noted for other cytokines (e.g., IL-10, IL-22, and TNF- α) in zebrafish and mammalian systems (e.g., Hernandez et al., 2018 *Science Immunol*; Gaublotte et al., 2015 *Cell*; Avraham-Davidi et al., 2023 *Nat Immunol*).

A relevant example is interferon lambda: despite their well-established role in gut inflammation, they are not detected in scRNA-seq datasets from the colon of DSS-treated mice (Mayassi et al., *Nature* 2024; link to dataset). Yet, their functional importance has been clearly demonstrated (e.g., Broggi et al., *Nature Immunology* 2017; Jena et al., *Cell* 2024).

It is also important to note that low mRNA levels do not preclude high or functionally relevant protein levels. We have tried to validate endogenous IL-26 protein levels by generating antibodies, but unfortunately, these efforts were not successful.

To address the reviewer's request for additional expression evidence, we have included new RT-qPCR data showing that *il26* is consistently expressed in dissected larval guts beginning at 3 dpf (line 102; Appendix Fig. S1B). To ensure specificity, we confirmed expression by running RT-PCR products on an agarose gel, which revealed appropriately sized spliced *il26* transcripts (Appendix Fig. S1A).

(2) Can the author also provide evidence to show whether IL-26 is also expressed in other immune cells besides ILCs? Within ILCs, is IL26 expressed in a specific ILC subgroup?

In the human gut, IL-26 is produced by adaptive and innate lymphocytes, with type 3 innate lymphoid cells (ILC3s) being the primary innate source (Hughes et al, 2009; Cella et al, 2009; Cols et al, 2016; Manel et al, 2008). Thus, IL-26 expression is generally restricted to the lymphoid lineage within immune cells. In addition, the adaptive immune system in zebrafish matures around 5 weeks of age (Lam et al, 2004). Therefore, this information combined with our observations presented in Fig. 8 support the conclusion that IL-26 is expressed by ILCs at the larval stage. However, to avoid overinterpretation, we do not claim that IL-26 is exclusively expressed by ILCs, nor do we mention its expression in other cell populations. We instead conclude that ILCs are the main contributors to *il26* expression in the zebrafish larval gut.

Regarding ILC subgroups, we have refined our scRNA-seq analysis (line 374; Figs. 8B,C). “To further characterize the identity of *il26*-expressing cells, we analyzed the expression of canonical ILC markers. While a small proportion of ILCs expressed ILC1 markers (*eomesa*, *tnfa*) and ILC2 markers (*gata3*, *il13*), a larger fraction expressed the ILC3 marker *rorc* (Fig. 8B). In addition, *il26*⁺ ILCs predominantly expressed *rorc*, while ILC1 and ILC2 markers were mostly undetected in these cells (Fig. 8C). These findings indicate that ILC3s are the main source of *il26* during inflammation in the larval gut.” However, to avoid overinterpretation—particularly extending conclusions beyond the inflammatory context—we have chosen to refer to the expressing population as “ILCs” rather than specifically “ILC3s” throughout the manuscript.

(3) Can the author also provide some phenotypes concerning the adult *il26*^{-/-} zebrafish to show the consequence of increased intestine epithelium cell proliferation and DNA damage observed in the embryonic stage? It is expected that the change in the gut microbial community and intestine epithelium cell phenotype during the embryonic stage due to the lack of *il26* may be exacerbated when the zebrafish continue to grow and develop. For example, is there an increased inflammation in *il26*^{-/-} adult zebrafish?

To answer to this concern, we quantified EdU and γ H2AX staining in 5-week post-fertilization (wpf) *il26*^{-/-} and WT fish (line 178). We chose this time point because the zebrafish immune system is morphologically and functionally mature around this period (Lam et al, 2004; Li et al, 2020; Stephens et al, 2016). Moreover, at this stage, the intestine undergoes key structural and functional changes, including the establishment of intestinal folds, increased complexity

of the epithelial architecture, and stabilization of the microbiota (Li et al, 2020; Stephens et al, 2016). No differences were observed in either EdU or γ H2AX numbers (Figs. EV4A-C). Additionally, we detected no differences in the expression levels of cell cycle markers (*myca*, *mycb*, *mych*; Figs. EV4D-F) or DNA repair genes (*brca2*, *p53*; Figs. EV4G,H). Moreover, expression levels of the cytokines *il1b*, *tnfa*, and *il10*, used to assess inflammation, were also comparable between *il26*^{-/-} and WT juvenile fish.

In the discussion in line 410, we argue that this could be due to the maturation of the adaptive immune system in juvenile zebrafish (Lam et al, 2004) or to feeding starting at 6 dpf in the animal facility. Profiling of the microbiota in juvenile and adult *il26*^{-/-} fish could clarify whether the dysbiosis observed in larval stages is resolved, potentially through adaptive immunity, or whether it persists but is mitigated by compensatory mechanisms that limit its impact on the gut epithelium.

(4) The authors indicate that the effects of *il26* on intestine epithelium proliferation and DNA damage are cell-type and intestine region-specific. It seems in most figures that the increased DNA damage was quantified in the very distal region of the intestine. However, in Figure 2, shows that DNA damage signal is mostly enriched in *fabp6* high cells. In zebrafish, *fabp6* is enriched in a small cell population that resembles the ileum (Wen J. et al., Science Advances, 2021), which is in the middle of the intestine. This seems to be contrary to the IF staining data. Can the author show staining evidence to further prove the intestine region-specific and the cell type-specific response induced by *il26*^{-/-}? In addition, can the author explain why such an intestine region/cell type-specific response is exhibited? For example, is there an enriched innate immune cell in the zebrafish ileum region in *il26*^{-/-}?

We sincerely thank the reviewer for this insightful comment and for drawing our attention to the specificity of *fabp6* expression. Upon revisiting our scRNA-seq dataset, we found that within the Absorptive_EC1 cluster, *fabp6*-low cells exhibit higher *tp53* expression compared to *fabp6*-high cells (Figure embedded below). This suggests the presence of transcriptionally distinct subpopulations within this cluster, and importantly, that the *fabp6*-high subset, associated with the ileum, does not appear to express high levels of DNA damage markers.

In line with this, we also observed that only a small proportion (9.7%) of Absorptive_EC1 cells express DNA damage markers such as *tp53*, supporting the idea that only a subset of this population is affected.

In light of these findings, we have removed the reference to *fabp6*-high cells from the revised manuscript and now rely solely on immunofluorescence imaging to delineate the region-specificity of DNA damage. This analysis consistently reveals increased γ H2AX signal in the distal region of the intestine in *il26*^{-/-} larvae (Fig. EV2C).

Regarding EdU staining, we have quantified EdU-positive cells across gut segments and found that *il26*^{-/-} larvae showed a statistically significant increase in the number of EdU-positive cells in the posterior gut. A similar, albeit statistically non-significant, trend was observed in the mid and anterior gut. (line 135; Figs. EV2A,B).

As for the underlying cause of this region-specific response, we believe it is microbiota-related rather than driven by the site of IL-26 production. IL-26 is a secreted cytokine and may diffuse across intestinal segments, exerting effects beyond its point of origin. Therefore, we hypothesize that the observed phenotypes could be explained by regionally distinct microbial communities along the intestinal tract that are differentially affected by the loss of IL-26. For example, in *il26*^{-/-} larvae, alterations in the posterior gut microbiota could lead to localized epithelial stress. Alternatively, epithelial cells in this region might be intrinsically more sensitive to microbial perturbations. Investigating these possibilities will be the focus of future research beyond the scope of the current study.

Minor concerns:

(1) Figure 1 E, both DNA and Edu are in blue color, making it extremely hard to see the Edu labeling. Consider changing the color of DNA to gray so that the other two labels can be visualized better.

We thank the reviewer for this helpful suggestion. We agree that the overlapping blue signals made it difficult to distinguish EdU labeling from DNA staining. In response, we have updated the image by converting the EdU channel to gray. This adjustment has also been applied consistently throughout the revised manuscript.

(2) Figure 1 E, please also add an image of the whole zebrafish intestine to show the intestine region-specific effects.

We have provided the requested images (Figs. EV2A,C).

(3) Some of the IF staining images, such as Figure S2A, are not clear. The γ H2AX is not visible in the image panels.

We thank the reviewer for pointing this out. We believe that the γ H2AX signal may have been obscured due to compression or resolution loss during PDF conversion. We have now provided high-resolution, production-quality images, and we hope this resolves the visibility issue.

Dear Pedro,

Thank you for submitting a revised version of your manuscript. We have now received input from two of the original reviewers, who are satisfied with the revisions and now recommend acceptance of the manuscript.

There now remain only a few editorial points that need to be addressed before I can extend official acceptance of the manuscript:

1. Please check that the funding information is correct and identical both in the manuscript and our online system. Currently, the PhD fellowships from the Ministère de l'Enseignement Supérieur et de la Recherche and from the Fondation pour la Recherche Médicale (FDT202304016654) are missing from our online system.
2. Please remove figures from the manuscript text file.
3. Please move "Data Availability" section to the end of "Methods" section.
4. CRedit has replaced the traditional author contributions section because it offers a systematic, machine-readable author contributions format that allows for more effective research assessment. Please remove the Authors Contributions from the manuscript and use the free text boxes beneath each contributing author's name in our online submission system to add specific details on the author's contribution. More information is available in our guide to authors.
5. In the Appendix, please add page numbers to the table of contents.
6. Figure panels 6G-H and 9A-C are not mentioned in the manuscript text. Please add the corresponding callouts.
7. In the "References" section, there is an incomplete reference to "Nanoporetech (2024) dorado". Please update with full information.
8. Please upload source data for Fig. 7D and 7G as excel files - we were not able to open it using our Prism Viewer application.
9. Our data editors have flagged the following issues in figure legends that need correcting:
 - Please provide the exact p values in the legends of figures 6F, 7D.
 - Please indicate the statistical test used for data analysis in the legends of figures EV1 A-D.
 - Please provide information on the number and nature of replicates in the legends of figures 4A, B.
10. Papers published in The EMBO Journal are accompanied online by a 'Synopsis' to enhance discoverability of the manuscript. It consists of A) a short (1-2 sentences) summary of the findings and their significance, B) 3-4 bullet points highlighting key results and C) a synopsis image that is 550x300-600 pixels large (width x height, jpeg or png format). You can either show a model or key data in the synopsis image. Please note that the image size is rather small and that text needs to be readable at the final size.

With best wishes,

Ieva

We realize that it is difficult to revise to a specific deadline. In the interest of protecting the conceptual advance provided by the work, we recommend a revision within 3 months (16th Nov 2025). Please discuss the revision progress ahead of this time with the editor if you require more time to complete the revisions.

Referee #2:

The authors have sufficiently addressed all of my concerns with new experimental data.

Referee #3:

The authors have addressed all my concerns. No other recommendation.

The authors addressed the remaining editorial issues.

Dear Pedro,

Thank you for addressing the final editorial issues. I sincerely apologise for the delay in the processing of your manuscript due to the currently high manuscript submission rate to our office. I am now pleased to inform you that your manuscript has been accepted for publication.

Before we will forward your manuscript for typesetting, I will look into the synopsis text that you kindly provided and will let you know if any textual edits to the journal style are needed.

If you have any questions, please do not hesitate to contact the Editorial Office. Thank you for this contribution to The EMBO Journal and congratulations on a nice study!

Best wishes,

Ieva
